# Evaluation of convection-permitting extreme precipitation simulations for the south of France

Linh N. Luu[1,2], Robert Vautard[1], Pascal Yiou[1], and Jean-Michel Soubeyroux[3]

[1]Laboratoire des Sciences du Climat et de l'Environnement, UMR 8212 CEA-CNRS-UVSQ, Université Paris-Saclay and IPSL, Gif-sur-Yvette, France
[2]now at Royal Netherlands Meteorological Institute (KNMI), De Bilt, Netherlands
[3]Météo-France, Toulouse, France

**Correspondence:** Linh N. Luu (linh.luu@knmi.nl)

**Abstract.** In the autumn, the French Mediterranean area is frequently exposed to heavy precipitation events whose daily accumulation can exceed 300 mm. One of the key processes contributing to these precipitation amounts is the deep convection, which can be resolved explicitly by state-or-the-art convection-permitting model to reproduce heavy rainfall events that are comparable to observations. This approach has been tested and performed at climate scale in several studies in recent decades for different areas. In this research, we investigate the added values of using an ensemble of three climate simulations at convection-permitting resolution (approx. 3 km) in replicating extreme precipitation events in both daily and shorter time scales over the South of France. These three convection-permitting simulations are performed with the Weather Research and Forecasting Model (WRF). They are forced by three EURO-CORDEX simulations, which are also run with WRF at the resolution of 0.11° (approx. 12 km). We found that a convection-permitting approach provides a more realistic representation of extreme daily and 3-hourly rainfall in comparison with EURO-CORDEX simulations. Their similarity with observations allows a use for climate change studies and its impacts.

## 1 Introduction

Deep convection is a key atmospheric process leading to heavy rainfall in a short duration that can generate floods and infrastructure destruction with a large impact on societies. This process has close interactions with other physical and micro-physical processes, large-scale and local dynamics of the atmosphere. However, deep convection processes have been parameterized in simulations at climate scale (i.e., more than ten years) for a long period of time. The parameterization methods that are based on statistical properties of convection processes within a grid box and their interactions with prognostic variables have been designed to represent this process at local scale (Kendon et al., 2012). This procedure brings large uncertainty to the results of climate models, including biases of rainfall characteristics such as the underestimation of short-duration extreme rainfall (Fosser et al., 2015; Hohenegger et al., 2008; Kendon et al., 2019; Lenderink and Meijgaard, 2008; Prein et al., 2013). Thanks to the rapid development of technology and computer power, a prominently emerging way that has been used in the recent two decades to resolve explicitly deep convection and avoid the application of convective parameterization schemes is to increase horizontal resolution to convection-permitting resolution (i.e., less than 4 km). Convection-permitting models hold promises

of representing central processes in the climate system and could make a step change in the climate projections as they better represent impactful precipitation extremes. There is also a hope that they could remove important biases if employed at global scale (Palmer and Stevens, 2019). However, this approach requires solving a trade-off between conducting several runs to generate large ensembles of simulations with a sufficient resolution (e.g., for impact of extreme event or attribution studies) and saving the expense of computing resource (Giorgi and Mearns, 1999; Trenberth, 2008).

Several simulations at convection-permitting resolutions were conducted for different regions and periods of time to examine the added value of this approach in reproducing precipitation (Prein et al., 2015). Regional models have been used to dynamically downscale the output of global models (e.g., CMIP5 (Taylor et al., 2012)) or reanalysis data (e.g., ERA-Interim (Dee et al., 2011)) to these convection-permitting resolutions. Generally, the simulated rainfall from those models have a better agreement with observations compared to those using convective parameterization schemes, especially in terms of temporal and spatial distributions of extreme rainfall event from sub-daily to daily time scales (Armon et al., 2019; Ban et al., 2014, 2018; Chan et al., 2013, 2014; Fosser et al., 2015; Hodnebrog et al., 2019; Kendon et al., 2012; Knist et al., 2018; Prein et al., 2013; Vergara-Temprado et al., 2021). Additionally, the diurnal cycle of precipitation is also better represented in this cloud-resolving model (Ban et al., 2014; Fosser et al., 2015; Knist et al., 2018; Langhans et al., 2013; Prein et al., 2013; Scaff et al., 2019). The increase of extreme hourly rainfall at super adiabatic rate (e.g., around 14%/°C) when scaling with temperature in observations analyses is also reproduced in convection-permitting simulations for a few areas (Ban et al., 2014; Knist et al., 2018). The convection-permitting models can also reduce the "drizzle problem" found in many lower-resolution simulations, which is characterized by frequent and persistent light rainfall events (Berthou et al., 2020; Fosser et al., 2015; Kendon et al., 2012; Lind et al., 2020). Scaff et al. (2019) additionally showed that the timing of the peak of convection was better performed by such very high resolutions. Given these added values of convection-permitting model in reproducing extreme precipitation events, a few studies used this approach for future projection of this variable with more confidence, especially in short duration events (e.g., Adinolfi et al., 2021; Broucke et al., 2019; Chan et al., 2020; Hodnebrog et al., 2019; Kendon et al., 2017, 2019; Pichelli et al., 2021).

The improvement in reproducing heavy precipitation events of the convection-permitting approach over parameterization methods comes from the higher resolution itself, the better representation of surface properties and complex topography (e.g., steep mountainous region), the explicit-solving of convection processes at local scale and its interaction with large-scale circulation and better solving of atmospheric dynamics (Feng et al., 2018; Prein et al., 2013, 2015). Despite those enhancements, the systematic errors are inevitable because of the use of other parameterization schemes and discrete numerical methods and the fact that convection (both deep and shallow) is not completely resolved at the grid spacing of 1 to 3 km (Broucke et al., 2019). For instance, a convection-permitting model could overestimate or underestimate heavy rainfall, prolong the duration of light rainfall, provide more extreme precipitation events than observations and fail in simulating the location of the heaviest rainfall events (Armon et al., 2019; Broucke et al., 2019; Chan et al., 2013, 2014; Fosser et al., 2015; Fumière et al., 2019).

The coastal regions along the Mediterranean frequently undergo very heavy precipitation events (e.g., hundreds of millimeters per day) in the autumn which subsequently lead to flash floods and landslides causing massive losses and damages (Delrieu et al., 2005; Fresnay et al., 2012; Llasat et al., 2013; Nuissier et al., 2008; Ricard et al., 2012). In addition, this area is considered

as a hotspot of climate change that strongly responds to warming at global scale (Giorgi, 2006; Tuel and Eltahir, 2020). As a result, the Mediterranean has received an increasing scientific interest in investigating the mechanisms leading to flood-inducing heavy precipitation as well as in improving the model ability to predict and project those events in a complex changing climate that provides substantial support to adaptation and mitigation for society (Drobinski et al., 2014; Ducrocq et al., 2014). The convection-permitting approach has been recently used to reproduce and obtain insights of extreme precipitation events in this area. Armon et al. (2019) showed that the convection-permitting model can reproduce the structure and location of 95% of 41 observed heavy precipitations events in the eastern Mediterranean and consequently suggested using this approach for a long simulation. Zittis et al. (2017) found that the convection-permitting model outperformed the convection parameterization approach for extreme rainfall events over the eastern Mediterranean. Fumière et al. (2019) used convection-permitting model to downscale the ERA-Interim reanalysis and proved that this high resolution improved the simulations of intensity and location of daily and sub-daily extreme precipitation. These results were confirmed over the area in a longer climate scale simulation by Caillaud et al. (2021). Berthou et al. (2020) also showed improvement of the convection-permitting model in reproducing daily heavy precipitation events in the autumn over the Mediterranean coasts at climate scale. Meredith et al. (2020) found that the convection-permitting model can improve sub-hourly intense precipitation. Coppola et al. (2018) used, for the first time, a multi-model approach with convection-permitting resolution to simulate a few case studies of heavy rainfall events over the European Mediterranean areas. They showed that each convection-permitting model can reproduce the case studies. However, the results among models spread when the event was convective and less constrained by the large-scale dynamics. This multi-model approach was then performed at climate scale with reanalysis forcing (Ban et al., 2021) and with CMIP5 forcing (Pichelli et al., 2021), which both found improvement in convective precipitation outcomes. These results highlighted the importance of using a multi-model approach to investigate convective precipitation events and stimulate the use of this approach in climate change impact study.

The assessment of convection-permitting simulations over long simulations requires further attention, in particular for the dynamical downscaling of global climate models used for projections of future scenarios. This article is designed to evaluate the skills of a unique regional climate model with a convection permitting set-up to simulate extreme hourly and daily precipitations in a region prone to such convective event in the Mediterranean area (Nuissier et al., 2008). The analysis made here is at a climate scale and is done by dynamically downscaling the climate information privded by the existing EURO-CORDEX experiments (Coppola et al., 2020; Jacob et al., 2014; Kotlarski et al., 2014; Vautard et al., 2020). The assessment is not made at the event or process levels but uses long simulations to evaluate whether the statistical properties (intensity, duration) of events are comparable to observations. The area under consideration in this article is the Cévennes mountain range, a part of the Massif Central in the south of France (1), where extreme precipitation events are most intense in France (Vautard et al., 2015). The experimental design of these runs, reference datasets and evaluation methods are presented in Section 2. The evaluation and discussion are given in Section 3. The last section presents the conclusion. We also provide supplementary information where a few simulations with convection-permitting configuration driven by the ERA-Interim are tested to select an appropriate domain.

## 2 Experimental design, data and methods

### 2.1 Experimental design

In this study, the Weather Research and Forecasting Model (WRF-ARW) version 3.8.1 is used to conduct several long simulations at convection-permitting resolution ($0.0275°$, approx. 3 km) for the French Mediterranean region. These simulations are forced by the three different EURO-CORDEX ($0.11°$, approx. 12 km, hereafter mentioned as EUR-11) simulations without nudging, for which outputs are available every 3 hours. These EUR-11 simulations were also done with WRF-ARW version 3.8.1 and driven by three GCMs including the IPSL-CM5A-MR, the HADGEM2-ES and the NORESM1-M (see Vautard et al.

(2020) for details about the new EURO-CORDEX ensemble). Each convection-permitting simulation (hereafter mentioned as CPS) is conducted for two different periods including 1951-1980 and 2001-2030 with the RCP8.5 scenario for the year after 2005. These two periods are chosen with a gap period (1981-2000) rather than a seamless one in order to perform an extreme event attribution study which will be presented in another article and needs a maximal time distance between two periods ("current climate" and "past climate").

The seasonal target of these downscaling experiments is the autumn when heavy rainfall events occur frequently over the Cévennes in the south of France. Hence, we initialize each autumn on August 26th and end the season on the 1st of December, as the extreme precipitation events in the Mediterranean coastal areas do not generally occur outside of this period and years can be considered independent from each other. Only a few days at the end of August are spent as spin-up time for the model to obtain the physical consistency among prognosis variables after being interpolated from 12 km to 3 km in the preparation

step, given that EUR-11 and CPS share the same regional model (i.e., WRFv3.8.1). Another factor that could take a long spin-up time (up to 10 years) in climate simulation is the soil moisture, especially the deep layer (Yang et al., 2011). Here, we facilitate our downscaling strategy (i.e. re-initializing the model every August) by interpolating the span-up soil moisture (and temperature) in EUR-11 results for the initialization of each season run of the CPS. Even though the imbalance of soil moisture is inevitable by doing so, this procedure is expected to minimize, to some extent, the perturbation in the land surface model of

WRF.

  The spatial configuration contains $301\times301$ grid points covering the Cévennes mountain range, a large part of the French Mediterranean region including Corsica (Figure 1). This size and position of CPS domain are selected after evaluating four different configurations for simulations of the autumn 2014 driven by the ERA-Interim reanalysis (see the Supplementary section). We use the same number of hybrid sigma vertical levels of 32 as the EUR-11 boundary conditions. The time step of

120 our simulations is 15 seconds, which is a fourth of the time step used in the EUR-11 (i.e. similar ratio as for resolutions). We adapt the same rotated map projection of EUR-11 simulations to these CPS experiments. Similar set of physics schemes as in EUR-11 simulations is used for these downscaling experiments. Those parameterization schemes incorporate the Thompson microphysics scheme (Thompson et al., 2008), the Rapid Radiative Transfer Model for GCMs (RRTMG) for long and short waves radiation (Iacono et al., 2008), the Monin-Obukhov (Janjic eta) surface layer scheme (Janjic, 1996), the Unified Noah

land-surface model and the MYNN scheme for boundary layer (Nakanishi and Niino, 2006). We also update the sea surface

temperature every day at midnight consistently to the EURO-CORDEX simulations (Vautard et al., 2020). The convection scheme is switched off in CPS simulations.

## 2.2 Evaluation methods

We use four indices to investigate the skills of CPS and EUR-11 simulations in reproducing extreme rainfall over the Cévennes mountain range. These indices consist of: i) comparing the autumn maximum rainfall (Rx); ii) comparing the distribution of wet events; iii) comparing the scaling of extreme precipitation and temperature at 2-meter height; iv) determining the total moisture source from the surface to 700 hPa (i.e. water vapour is mainly concentrated this low level of the atmosphere) that transports from the Mediterranean to the Cévennes.

The first index takes the average of autumn maxima rainfall values for the considering period. The second index compares the cumulative distributions of all rainfall values that are greater or equal to 0.1 mm (defined as wet events) in the considered period. The third index determines the non-parametric scaling of extreme precipitation with the increase in surface temperature proposed by Lenderink and Meijgaard (2008). In particular, we pair the rainfall (daily or hourly) dataset with their corresponding daily mean 2m temperature over the Cévennes. This pairing dataset is then sorted in the ascending order of temperature. Next, we divide this dataset into several bins whose width is 2°C with 1°C overlapping between the two consecutive bins and calculate the 99th percentile for rainfall and the mean temperature for each bin. We use a threshold of having at least 300 points of precipitation to take a bin into consideration. This is to avoid the under-sampling effect on the final scaling results (Boessenkool et al., 2017). For making inference for each bin, we use a non-parametric bootstrap by picking 1000 samples of pairing temperature and rainfall with replacement from that original bin. Each sample size is the same as that of the sample of the original bin. We repeat calculating the statistics for each bin and then estimate the 90% confidence interval of each bin based on those 1000 samples. The first three comparisons are applied to both daily rainfall events and daily maximum of 3-hour rainfall events which create six indices.

An important ingredient facilitating the mechanism of the severe precipitation events over the Cévennes in the autumn is the abundance of moisture from the Mediterranean, which is enhanced by a warm sea surface being exposed to the heat during the summer. This moisture is conveyed by unstable low-level southeastern flows produced by the usual development of a synoptic-scale trough to the west of the region during this season. This massive amount of moisture toward the Cévennes is then forced to lift up by high and steep orography of that triggers the quasi-stationary mesoscale convective system over the area. The updraft in this system is frequently strengthened at the same location as long as the low-level moist flows are persistent and intensified (Ducrocq et al., 2008, 2014; Lebeaupin et al., 2006; Lee et al., 2018; Nuissier et al., 2008, 2011). Based on these features, we propose a the fourth index to investigate the model ability in producing this low-level moisture transport impinging on the Cévennes mountain range using the method in Lélé et al. (2015):

$$\overrightarrow{Q} = -\frac{1}{g} \int\limits_{ps}^{pu} \overrightarrow{q\overrightarrow{U}}\, dp \tag{1}$$

In this equation, $\overrightarrow{Q}$ is the horizontal moisture transport vector (kg.m$^{-1}$.s$^{-1}$), $g$ is the standard gravitational acceleration at mean sea level (9.81 m.s$^{-2}$), $q$ is water vapour content (kg/kg), $\overrightarrow{U}$ is zonal and meridional wind vector (m.s$^{-1}$), $ps$ and $pu$ are surface and upper pressure level, in this case, 1000 hPa and 700 hPa, respectively. This equation is used to estimate the moisture transport for the 12 heaviest daily rainfall events occurring over the Cévennes mountain range in 30 years of each simulation and observations. Before selecting the events in each dataset, we first define the Cévennes box (see 1) by deriving the maxima and minima of the latitudes and longitudes of the 14 stations along the Cévennes used in Vautard et al. (2015). This box is roughly limited from 2.6°E to 5°E and from 43.3°N to 45.1°N. Next, we determine the 12 maxima rainfall values and their date of occurrence at any station within the box. We extract the maxima from all stations (i.e., not only considering the 14 stations) we have as long as those stations are located within the box. We eliminate a less heavy event between the two events occurring within 7 days that are both ranked on the top 12 heaviest events, so that we can avoid considering twice the same large-scale dynamic and moisture characteristics leading to those heavy rainfall events. For each determined event, we take the average of a few time steps (every 3 hours for simulations and every 6 hours for benchmark dataset) from 18 UTC of the previous day to 21 UTC of the day that event happened. Finally, we compute the mean moisture transport of those mean values of the 12 heaviest rainfall events over the Cévennes box.

## 2.3 Reference datasets

In this study, we use four reference datasets to evaluate the CPS and EUR-11 simulations. The first dataset includes in situ observations of daily and daily maximum of 3-hour rainfall. The daily rainfall data spans 1961 to 2014. The sub-daily dataset is available from 1982 (a few stations start only from 1998) to 2018. The second reference dataset is the SAFRAN reanalysis (Quintana-Seguí et al., 2008; Vidal et al., 2010). SAFRAN is provided in an hourly interval with the horizontal resolution of 8 km and start from 1958. We only use this dataset to evaluate the daily precipitation event because the hourly rainfall was interpolated from daily data that made its quality insufficient (Vidal et al., 2010). The third reference dataset is the COmbinaison en vue de la Meilleure Estimation de la Précipitation HOraiRE (COMEPHORE), which is a combined product of rain gauge and radar observations (Tabary et al., 2012). This dataset has high temporal (1 hour) and spatial (1 km) resolutions and higher quality than any other gridded observations in France, especially over the complex terrain regions (e.g., the Cévennes, Fumière et al. (2019)). However, this dataset only covers 11 years, from 1997 to 2007. The last dataset used for the evaluation is the ERA5 atmospheric reanalysis (Hersbach et al., 2020), which is a new-released dataset to replace the ERA-Interim operationally stopped in 2019. The ERA5 has a higher horizontal resolution (approx. 38 km) compared to its predecessor ERA-Interim and starts from 1979. We collect a few variables in pressure level such as the horizontal winds and specific humidity to serve the moisture transport investigation. Given that the time span of these reference datasets is different, we select different periods of simulations to be evaluated using different indices proposed in the beginning of this section. The selection of periods of simulations and reference datasets corresponding to each index is described in Table 1 below.

## 3  Evaluation results and discussion

In this section, we analyse and discuss the performance of EUR-11 and CPS simulations following the indices proposed in Section 2.2. For the Rx1day and Rx3hour, we upscale the CPSs and COMEPHORE data to the coarsest resolution (i.e., 0.11° of EUR-11) using a conservative remapping method (Jones, 1999) to broaden our discussion of the added value of the CPSs. Specifically, we evaluate the capability of CPSs in reproducing local features against in situ observations and the original COMEPHORE data, while we analyse the added value of CPSs by comparing the upscaled CPSs (denoted hereafter as CPS-11s) and EUR-11 to SAFRAN and upscaled COMEPHORE datasets. For those climate impact-oriented indices such as the distribution of wet events and the scaling of extreme precipitation with temperature, we only compare model results to in situ observations. Given the fact that EUR-11 and CPS simulations share the same regional model (i.e., WRF-ARW version 3.8.1) and physics, we mention each simulation shortly by its resolution combining with the driving GCMs (e.g., EUR-11-IPSL-CM5A-MR).

### 3.1  Autumn maximum daily rainfall (Rx1day)

We first look at the spatial distribution of the mean of autumn maxima daily rainfall (Rx1day) from all simulations and observations (Figure 2). The results from the two reference datasets including the SAFRAN (Figure 2-j) and the in situ observations (Figure 2-k) show that daily rainfall events occur along the Cévennes mountain range (i.e. the diagonal of the Cévennes box), especially over its northern part (i.e. above the latitude of 44°N). The maximum and mean of 14 stations (as used in Vautard et al., 2015) from the SAFRAN and observations are close to each other, 98 mm and 77 mm, respectively for SAFRAN versus 97 mm and 81 mm for observations. This coherence comes from the fact that SAFRAN is an interpolation product from in situ observations.

Generally, we observe an agreement between all simulations and reference datasets that rainfall patterns are heavier along the Cévennes. However, the intensity of Rx1day from the CPSs (and CPS-11s) and their driving EUR-11 are very different. The three EUR-11 simulations (Figure 2-a-c) show large dry biases over the Cévennes box. The mean dry biases over the box from those simulations range from 20% (EUR-11-HadGEM2-ES) to 39% (EUR-11-NorESM1-M) in comparison with SAFRAN. The two CPS-11s slightly underestimate Rx1day over the Cévennes box with a dry bias ranging from 7% (CPS-11-IPSL-CM5A-MR) to 20% (CPS-11-NorESM1-M), while the CPS-11-HadGEM2-ES rather overestimates Rx1day by 12% (Figure 2-g-i). For the CPS (Figure 2-d-f), all simulations underestimate Rx1day over the Cévennes box by -38% to -14% compared to in situ observations. In contrast, all simulations tend to show a wet bias in Rx1day over the French Alps. The wet biases in this area are more intensified by the CPSs. We find that the behaviour of CPSs depends on their driving EUR-11 simulations. The EUR-11-HadGEM2-ES or CPS- HadGEM2-ES shows the best agreement with observations when comparing them with other simulations with the same resolution.

## 3.2 Autumn maximum of daily maximum 3-hours rainfall (Rx3hour)

The convection-permitting model is expected to improve the representation of the deep convection process that leads to heavy
precipitation at local scale in a short period of time (e.g., sub-daily time scale). In this section, we investigate the autumn maximum 3-hour rainfall (Rx3hour) to clarify how much the CPSs could improve the short-duration rainfall in comparison with their driving EUR-11 simulations for the period of 2001–2030. We use the COMEPHORE (1997-2007), upscaled COMEPHORE and rain gauge measurement (1998-2018) for this evaluation. We skip the SAFRAN dataset due to its insufficient quality of hourly rainfall, which was obtained by interpolation process from daily data in combination with analysed hourly specific humidity and other factors (Vidal et al., 2010).

The spatial distributions of Rx3hour from simulations and observations are shown in Figure 3. We find that heavier rainfall events are still observed along a northeast-southwest axis, as for Rx1day. In addition, this pattern is expanded to the plain area on the south-east of the Cévennes range in the 11-years mean of COMEPHORE (Figure 3-j) and 21-years means of in situ observations (Figure 3-l). The spatial max/mean of Rx3hour of all 23 stations located within the Cévennes box from COMEPHORE are 81mm/45mm. The mean value of those 23 stations is consistent with the mean values from in-situ observations, but the maximum value is almost 30% larger compared to those from rain gauge data. This discrepancy could be explained by either the method applied to combining radar and in situ observations or the uncertainty in radar information over a complex topography area despite the good coverage of the radar system.

All simulations reproduce this coverage pattern of Rx3hour well, despite the fact that their magnitudes of the event vary compared to the observations. This is consistent with what was found by those analyses of Fumière et al. (2019) who estimated the extreme tail percentile, rather than the mean, of daily and hourly rainfall from convection-permitting driven by reanalysis ERA-Interim data. As expected, the EUR-11 simulations underestimate 3-hour extreme rainfall over the Cévennes box. The mean dry biases of Rx3hour over the Cévennes box from EUR-11-IPSL-CM5A-MR, EUR-11-HadGEM2-ES and EUR-11-NorESM1-M against upscaled COMEPHORE are -55%, -52% and -56%, respectively (Figure 3-a-c). The results from CPS-11s (Figure 3-g-i) also underestimate the extreme from upscaled COMEPHORE. Their spatial mean precipitation biases over the Cévennes box range from -18% (CPS-11-NorESM1-M) to -1% (CPS-11-HadGEM2-ES). These CPS-11s alos perform better than the EUR-11 in reproducing heavy rainfall over the plain and coastal area to the east of the Cévennes mountain range and over the Alps. For the CPSs, we find dry biases of those simulations compared to in situ observations. The mean biases of 23 stations within the Cévennes box from the CPSs range from -23% to -37% (Figure 3-d-f). In summary, the convection-permitting model show consistent skills and improves the reproduction of spatial distribution of heavy rainfall from daily to sub-daily time scale. We also find the coherence in the results in CPS and its driving EUR-11 simulations (e.g., those simulations from HadGEM2-ES experiments have better performance compared to others).

## 3.3 Distribution of wet events

In this section, we compare the station-pooling distributions of wet events (3-hourly/daily amount >= 0.1 mm) and the biases of the right tail (10%) of those distributions from all simulations including the upscaled CPS (i.e., CPS-11) against the in situ

observations. The results from the CPS-11 are close to those from the CPS (around 5% discrepancy similarly to what is shown in the previous sections) for both daily and 3-hourly rainfall. This enables us to find the advantage of CPSs in simulating the extreme rainfall events compared to the EUR-11 simulations. The analysis for daily rainfall is shown in Figure 4. In general, the tail of daily rainfall events is underestimated in all simulations. However, the CPS-11s (and hence the CPSs) show better agreement with in situ observations. Their mean biases in the 10% tail range from -45% to -20%. The dry mean biases of EUR-11 simulations range from -60% to -50% for the 10% right tail of the distributions (Figure 4-b). The improvement in reproducing the extreme event of CPSs compared to EUR-11s is more obvious in the analysis of 3-hourly events (Figure 5). The distributions of 3-hourly wet events from the CPS-11s and CPSs are close to in situ observations (Figure 5-a). The mean biases in the right tail of these simulations range from -25% to 10%. The dry mean biases of EUR-11 simulations remain similar to their analysis of daily wet events (approx. -65%) (Figure 5-b). For either daily or sub-daily wet events, we find that the downscaling experiments from the HadGEM2-ES achieve the best skills in reproducing extreme rainfall events in comparison with other simulations with corresponding resolutions.

## 3.4   Scaling extreme rainfall with surface temperature

From the Clausius-Clapeyron relation, we can infer that when the atmospheric temperature increases by 1 K (or °C), the capacity of the atmosphere in holding water vapour accordingly increases by approximately 7%. This means that given the absence of significant changes in relative humidity, the water vapour supplied for the convection may increase following the Clausius-Clapeyron relation when the atmospheric temperature increases (Lenderink and Attema, 2015). This relationship links the increase in extreme daily and sub-daily time scale to regional and global warming (Lenderink et al., 2017; Pall et al., 2007; Westra et al., 2014). In this section, we model the relation between extreme precipitation and daily mean surface temperature, which is theoretically reflected by the Clausius-Clapeyron relation, by a simple non-parametric scaling method described in section 2.2. We apply this method to EUR-11 and CPS simulations and then compare to the result obtained on in situ observations. We use 14 stations as in Vautard et al. (2015) and Luu et al. (2018) for the scaling of daily rainfall and 23 stations within the Cévennes box for the scaling of 3-hourly rainfall.

Figure 6-a compares the scaling model of extreme daily precipitation ($99^{th}$ percentile) with daily mean surface temperature from all simulations against in situ observations. The analysis of observations (black line) over the Cévennes shows that the dependence of extreme rainfall on the increasing in surface temperature closely follows the Clausius-Clapeyron (C-C) relation (black dotted lines in Figure 6) for the temperature above 2°C and breaks once exceeding 13°C. The scaling behavior of each CPS replicates its driving EUR-11 simulation for the daily precipitation scaling analysis, but the rainfall intensity from CPSs is higher. Specifically, the 2 downscaling simulations of the IPSL-CM5A-MR reproduce roughly the C-C relation in a range of 9°C to 17°C, while the 2 downscaling simulations of the HadGEM2-ES follow the C-C relation in range of 5°C to 13°C. The 2 simulations of NorESM1-M show similar behavior that follows the C-C relation in a range of roughly 4°C to 14°C. The overall scaling rate from EUR-11 simulations are close to observations, while CPSs slightly overestimate this rate.

The analysis for scaling of extreme 3-hourly rainfall with daily mean surface temperature is presented in Figure 6-b. We show that the observations analysis follows the super C-C relation for the temperature range of 6°C to 13°C. This super

relation can be directly explained that the latent heat released during the condensation period of water vapor can enhances the moisture convergence in lower level and the cloud dynamics (Trenberth et al., 2003; Lenderink et al., 2017). However, this result is different from what was found in (Drobinski et al., 2016). Their analysis showed that this scaling follows the Clausius-Clapeyron relation rather than the super Clausius-Clapeyron relation. This difference could come from the fact that Drobinski et al. (2016) used more than 200 stations which cover a large area in the South of France (i.e., not only restricted to

the Cévennes) and they did not focus only on the autumn. This leads to the mixture of different patterns of rainfall in different seasons and areas.

For the simulations, we find that CPSs can reproduce the super C-C relation similarly to observations (Figure 6-b). The CPS-IPSL-CM5A-MR shows a super C-C scaling in the range of 9°C to 17°C. The CPS-HadGEM2-ES and CPS-NorESM1-M follow super C-C in the range of 5°C to 17°C and 7°C to 14°C, respectively. These simulations also have better agreement

with observations in terms of intensity. In contrast, the three EUR-11 simulations are unsuccessful in approximating the super scaling behaviour and especially the rainfall intensity. We explain this underestimation by the fact that the resolution of EUR-11 is insufficient to reproduce the more localized extreme events and that the convection scheme and that the convection scheme used in EUR-11 over-simplified the cloud process by statistical distributions and imposing assumptions of quasi-equilibrium with large-scale forcing (from grid points), approximation of moist air entraining in the updraft, and representation of all single

cloud elements by sole steady state updraft of the whole cloud ensemble (Houze, 2004; Lenderink and Attema, 2015; Prein et al., 2013; de Rooy et al., 2013). In addition, we find the decreasing trend (i.e., the hook shape) of this scaling model in high temperature ranges for both daily and sub-daily precipitation. Because we use surface temperature as a proxy of condensation temperature (i.e., dew point), we overestimate the real saturation temperature (Drobinski et al., 2016). In other word, this hook shape results from the lack of sufficient water vapour in the atmosphere (Jones et al., 2010), therefore the condition of saturation

is broken.

### 3.5 Moisture sources

In this section, we investigate the ability of the model in reproducing the mean moisture source brought by the south-eastern flow impinging on the Cévennes. We use the method described in Section 2.2 to compare the mean moisture transport of the 12 heaviest Cévennes events in each simulation against the ERA5 reanalysis.

The comparison of the mean moisture transport of the 12 heaviest Cévennes rainfall events occurring over the Cévennes box is shown in Figure 7. Because the size of CPS domain is insufficient for this large-scale analysis, we visually embed each CPS domain inside its driving EUR-11 domain in each panel showing the results from CPSs (Figure 7-d-f). This means that we estimate the mean moisture transport of the 12 heaviest Cévennes events from each CPS simulation and the corresponding information from its driving EUR-11 to perform those plots. Therefore, the information from EUR-11 simulations in those

cases (Figure 7-d-f) may differ from those mean moisture transport investigations for the 12 heavy Cévennes events determined from EUR-11s themselves (Figure 7-a-c). The result from the ERA5 reanalysis indicates a low-pressure system locating around 50N and 9W in the north Atlantic with its trough expanding to the south (Figure 7-g). This large-scale system produces southerly to easterly flows that transport the moisture from the warm Mediterranean hitting the Cévennes. The mean moist flux

covering the Cévennes box in this case is 265 kg.m$^{-1}$.s$^{-1}$ and larger than surrounding areas. All EUR-11 simulations (i.e. either analysis of themselves in Figure 7-a-c or complement large-scale dynamics information for the CPS analyses in Figure 7-d-f) can reproduce well these synoptic features. The low-pressure systems are generally located between 45°N to 50°N and 5°W to 10°W. These systems enable the low-level flows bringing larger water vapour content into the Cévennes box compared to nearby areas in all simulations in a way that is coherent with ERA-5 analysis. The bias of mean moisture source on the Cévennes box from EUR-11 simulations is roughly 25% lower than in ERA5. The CPS simulations can reproduce better agreement of moisture source over the Cévennes box with ERA5, in spite of their restriction in domain size. The mean moisture of the 12 heaviest rainfall events over the Cévennes box from CPSs are approximately underestimated by 17% compared to the ERA5. In summary, all simulations can reproduce the moisture source hitting the Cévennes, with a slightly better performance from CPSs. However, the CPSs show more added values in reproducing more realistic extreme precipitation events. This suggests that the explicitly-resolving convection, finer resolution and more elaborated topography all play a role in this improvement.

## 4  Discussion and conclusion

In this study, we conduct three dynamical downscaling experiments from 12 km to 3 km using the WRF-ARW version 3.8.1 for two different periods including 1951-1980 and 2001-2030. These simulations, following a few experiments of 3-month simulations driven by ERA-Interim for testing and selecting an appropriate configuration, are driven by the three EURO-CORDEX simulations using the same WRF-ARW version which downscaled three GCMs from CMIP5 including IPSL-CM5A-MR, HADGEM2-ES and NORESM1-M. We simulate precipitation only for the autumn over the French Mediterranean with focusing on the South of France. This downscaling strategy benefits from time and energy efficiency that we can run simulations for different autumns and experiments at the same time.

We find that convection-permitting simulations (CPS(s)) can reproduce more realistic heavy precipitation events in terms of magnitude, spatial coverage and statistical properties than EURO-CORDEX simulations. This improvement is more pronounced in 3-hourly rainfall analysis than in the daily one. These features are robust at both climate scale driven by EUR-11s and the 3-months scale driven by the reanalysis (see the supplementary) to reproduce a few specific events occurring in the autumn 2014. In addition, the CPSs at climate scale can reproduce a double of the rate of the Clausius-Clapeyron relation for the scaling of 3-hourly rainfall to surface temperature that is absent in the EUR-11 simulations with convection parameterized method and reproduce the high extremes in plain areas. These findings are coherent with other studies with a convection-permitting approach forced by reanalysis data for the autumn events over the French Mediterranean region (Berthou et al., 2020; Caillaud et al., 2021; Fumière et al., 2019; Lenderink et al., 2019) and lately forced by CMIP5 models (Pichelli et al., 2021), and for other seasons and areas (Armon et al., 2019; Ban et al., 2018, 2021; Kendon et al., 2012; Knist et al., 2018). We also note that our findings remain similar when all results from the CPSs are upscaled to match the resolution of the EUR-11 that is usually applied in high-resolution model evaluation by several studies.

We also find that the behaviour of CPS simulations is modulated by their driving GCM simulations given that they share the same regional climate model (e.g., WRF model). The biases of the driving GCMs can be conveyed into the EUR-11s, and hence

to the CPS simulations. For example, the downscaling experiment of the HadGEM2-ES show the best performance compared to others at the same resolution, while those from NorESM1-M show larger dry biases compared to the rest. We have verified that the bias in sea surface temperature (SST) over the French Mediterranean region during the 12 heaviest precipitation events was over 2°C underestimated by the NorESM1-M, while the others showed slightly overestimation (figure not shown here). The decrease in SST weakens the convection, hence potentially affecting the extreme precipitation (Lebeaupin et al., 2006). This emphasizes the role of boundary conditions on the feedback of nested domains.

Both EUR-11 and CPS simulations can reproduce the moisture transport hitting the Cévennes with slightly better agreement of CPS with ERA5 in terms of mean amount of moisture on the Cévennes box. Even though the moisture source is well presented in all simulations, with a slight enhancement in CPS simulations, only three CPS simulations are able to reproduce realistic sub-daily extreme precipitation over the Cévennes. It can be deduced that convection-permitting features, higher resolution, better representation of complex orography and a better supply of moisture source can all play a role in the added values of convection-permitting simulations.

One of the remaining inherent problems in evaluating long simulations at hourly time scale is the uncertainty in observations (as mentioned in Ban et al., 2014, 2021). The 3-hourly observational dataset used in this research started at different times among stations. In addition, the coverage of stations, especially inside the Cévennes box, is limited only to the south-eastern part of the area. A large part in the north of Cévennes range where a lot of heavy rainfall value happened is missing (as shown by COMEPHORE data). The COMEPHORE data, which is the combination of radar measurement and in situ observations, provides a better representation of the spatial distribution of heavy rainfall. Even though this data also contains a lot of uncertainty which comes from poor observations and radar information over the complex topography, its quality over the Cévennes is sufficient (as discussed in Fumière et al., 2019). However, the length of this dataset is quite short and its observation period is different from the simulations in this research.

We conclude that a convection-permitting approach with the WRF regional climate model appears to provide a fairly realistic representation of extreme daily and 3-hourly rainfall simulations. Their similarity with observations allows for a use for climate change studies and their impacts. They should provide more reliable simulations than GCMs or even the high-resolution EURO-CORDEX simulations. However, we suggest using a multi-model approach to have a better consideration in the sensitivity of this variable on different model dynamics or micro-physic schemes (Ban et al., 2021; Pichelli et al., 2021).

*Data availability.* The ERA5 reanalysis data can be found at Copernicus Climate Data Store (https://cds.climate.copernicus.eu) and the EURO-CORDEX simulations can be found at https://esgf-node.ipsl.upmc.fr/search/cordex-ipsl/

*Author contributions.* LL set up and ran the convection-permitting simulations, designed the article, produced all figured and wrote the main text. J-MS provided the 3-hourly in situ observations. All authors contribute to the review and writing.

*Competing interests.* We declare that there is no competing interest.

*Acknowledgements.* LL was supported by the Commissariat à l'Energie Atomique et aux énergies alternatives (CEA). This work was supported by an ERC grant No. 338965-A2C2, and the EUPHEME project, which is part of ERA4CS, an ERA-NET initiated by JPI Climate and co-funded by the European Union (Grant No. 690462). This work is also part of the Convention on financial support for climate services supported by the French Ministry for an Ecological and Solidary Transition.

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

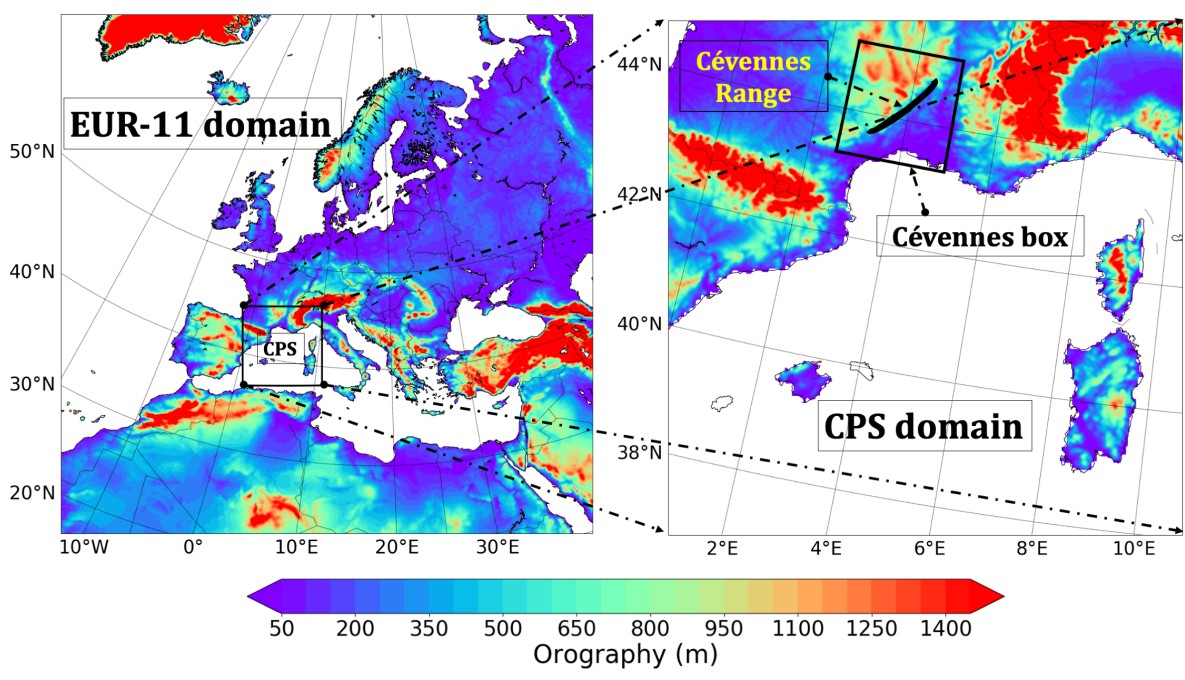

**Figure 1.** The domains of EURO-CORDEX and convection-permitting simulations. The shading colours denote the surface height above mean sea level from WRF.

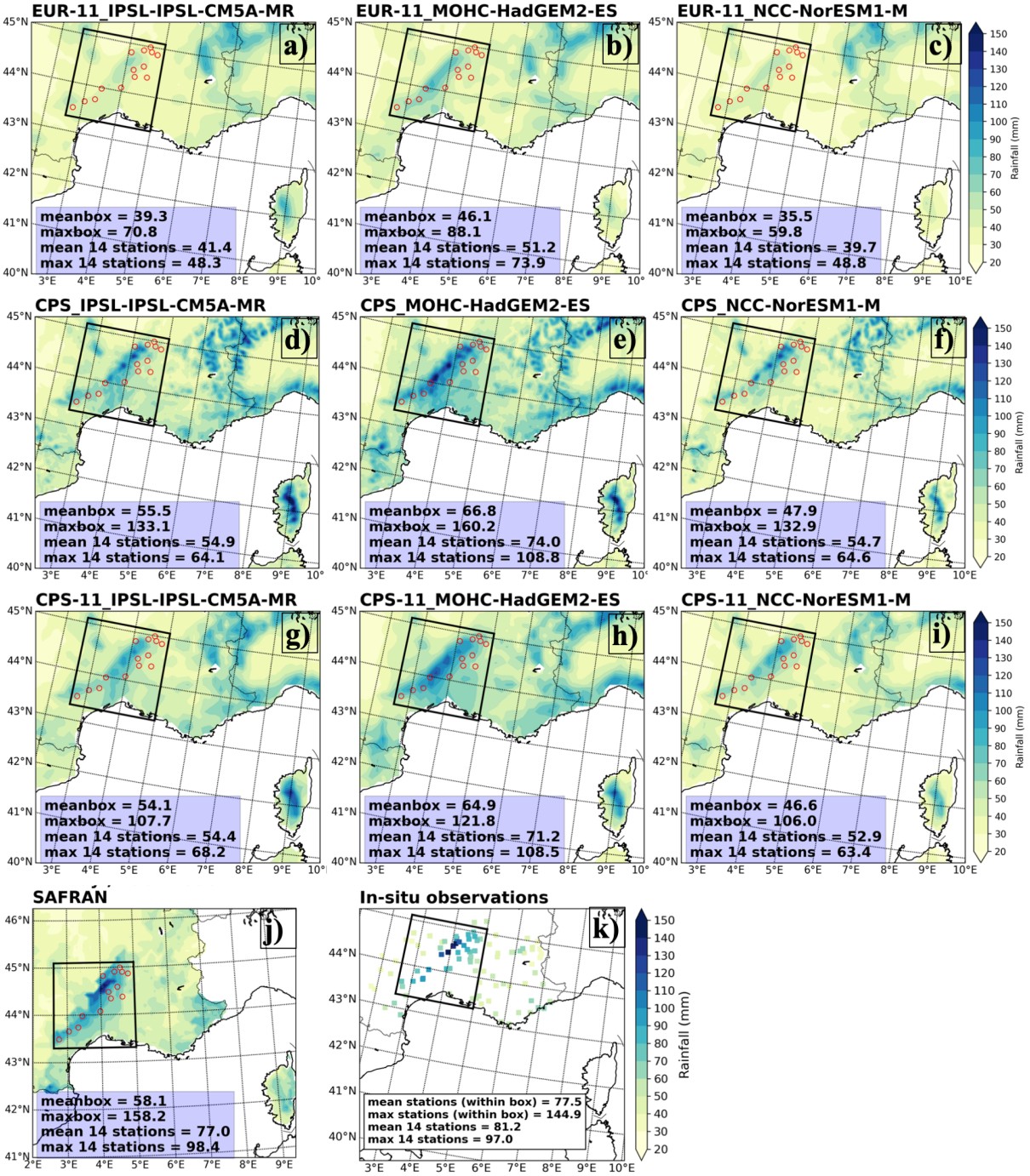

**Figure 2.** The Autumn maximum daily rainfall (Rx1day) from EUR-11 (a-c) simulations (1951–1980), CPS (d-f) simulations (1951-1980), CPS-11 (g-i), SAFRAN (j) (1961-1990) and in situ observations (k) (1961-1990) data; The red empty circles inside the Cévennes box from panel a to j denote 14 stations used in Vautard et al. (2015)

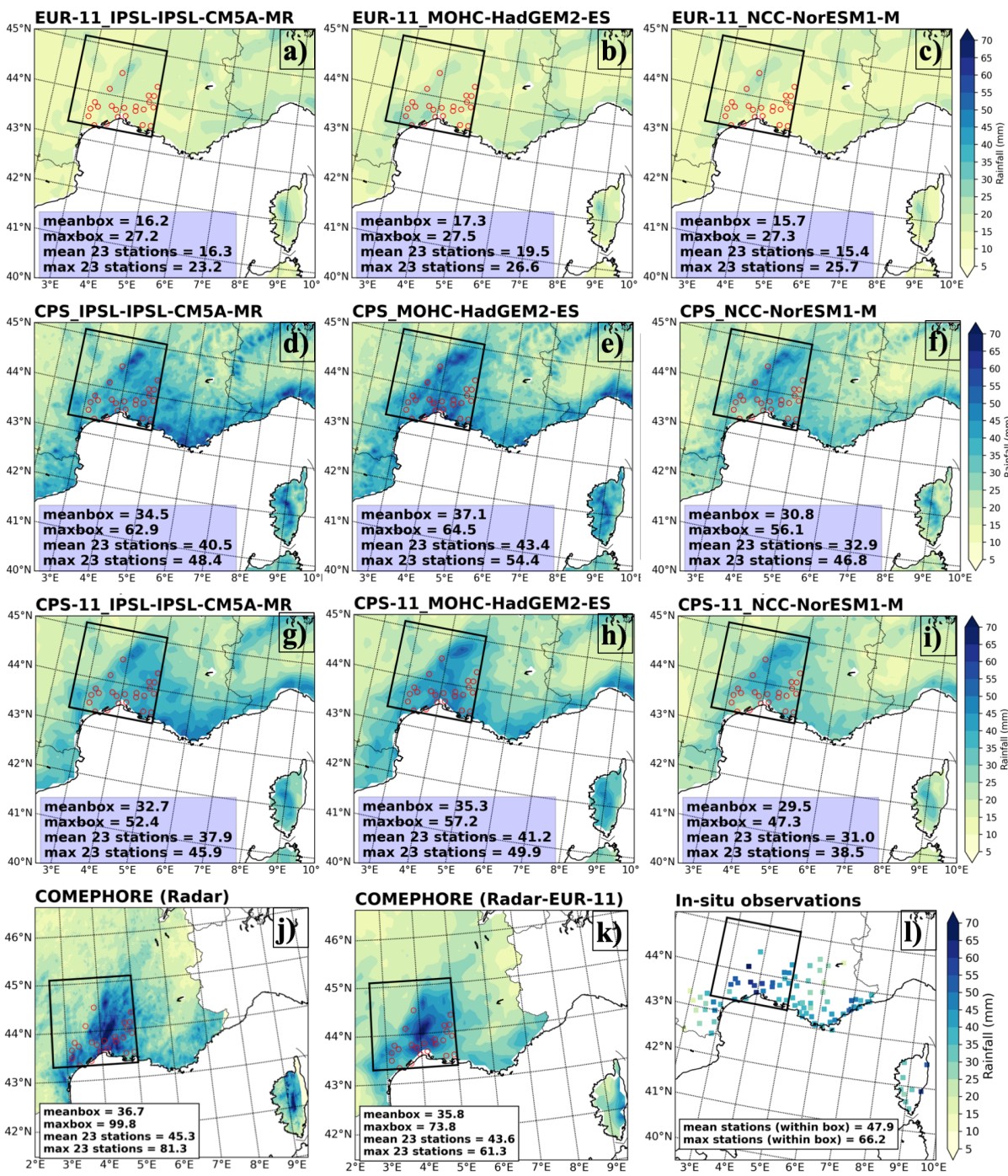

**Figure 3.** The Autumn maximum 3-hour rainfall (Rx3hour) from EUR-11 (a-c) simulations (2001-2030), CPS (d-f) simulations (2001-2030), CPS-11 (g-i), COMEPHORE (j) dataset (1997-2007), upscaling COMEPHORE (k) and in situ observations (l) datasets (1998-2018); The red empty circles inside the Cévennes box from panel a to k denote 23 stations that 3-hourly data is available.

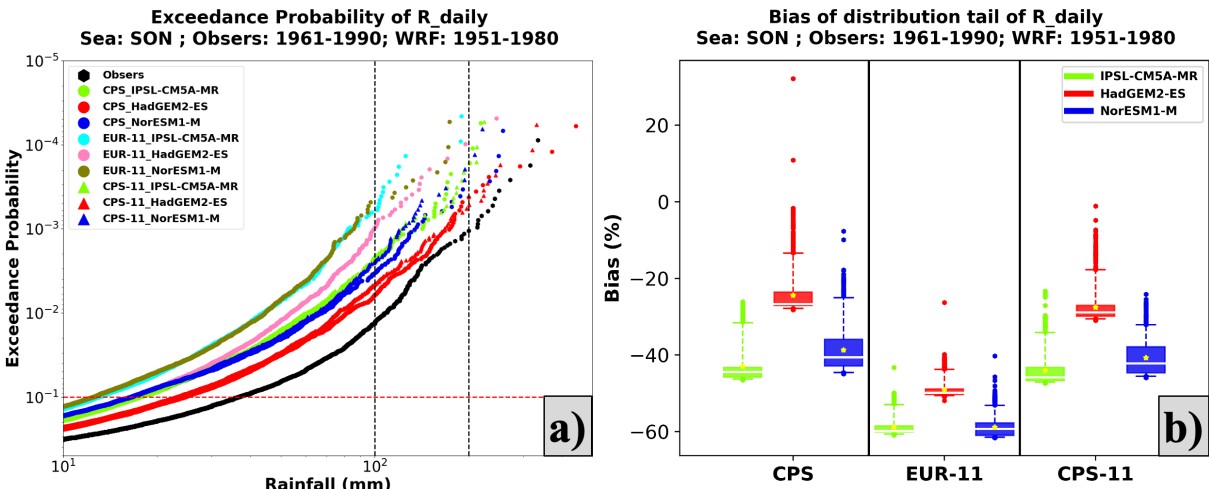

**Figure 4.** Exceedance probability distribution (a) for daily rainfall in the autumn from in situ observations (1961-1990) and all simulations (1951-1980) and the bias (b) of 10% in the tail of the distribution from each simulation against in situ observations. The red dotted line on panel a denotes the exceedance probability of 0.1 above which the simulated rainfall values are used to estimate the bias of the distribution tail on panel b.

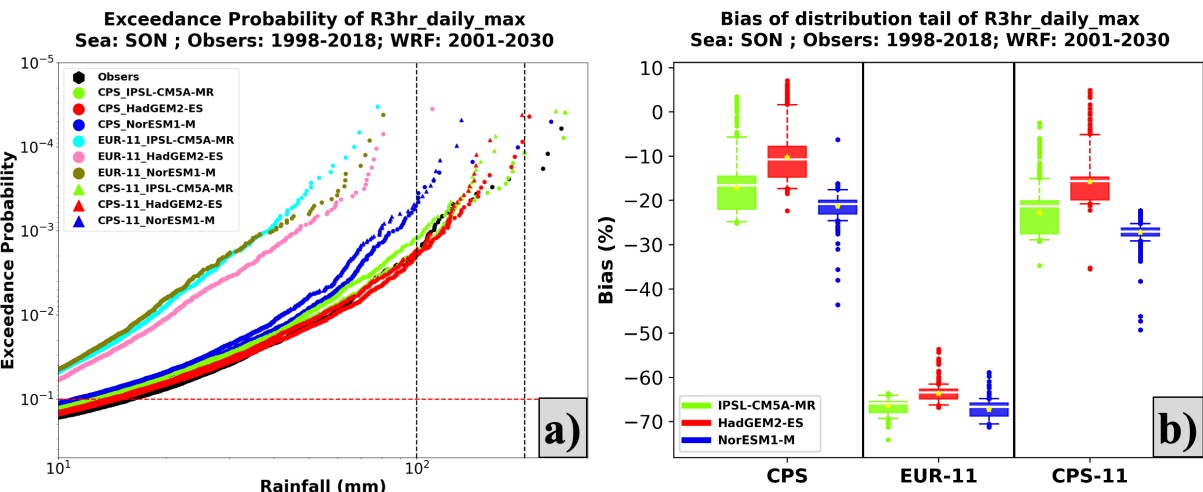

**Figure 5.** Exceedance probability distribution (a) for daily maximum of 3-hour rainfall in the autumn from in situ observations (1998-2018) and all simulations (2001-2030) and the bias (b) of 10% in the tail of the distribution from each simulation against in situ observations. The red dotted line on panel a denotes the exceedance probability of 0.1 above which the simulated rainfall values are used to estimate the bias of the distribution tail on panel b.

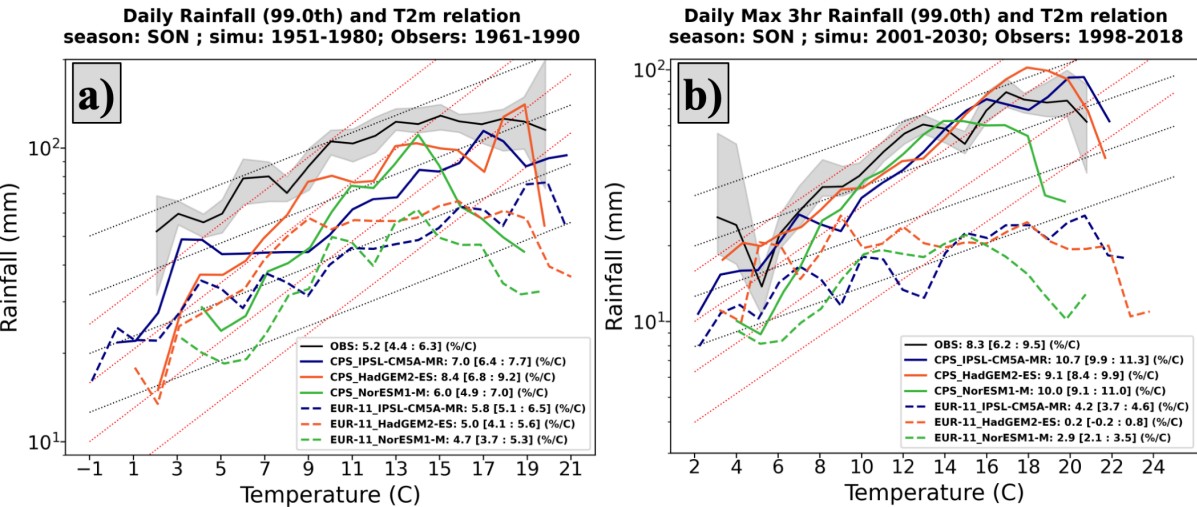

**Figure 6.** Extreme (99th percentile) daily precipitation (a) and daily maximum of 3-hourly rainfall (b) in scaling with daily temperature at 2m from simulations (1951-1980 for daily rainfall and 2001-2030 for 3-hourly rainfall) and in situ observations (1961-1990 for daily rainfall and 1998-2018 for 3-hourly rainfall); the black dot lines show Clausius-Clapeyron relation and the red dot lines show the super Clausius-Clapeyron relation; the grey band denotes 90% confident interval of observational scaling.

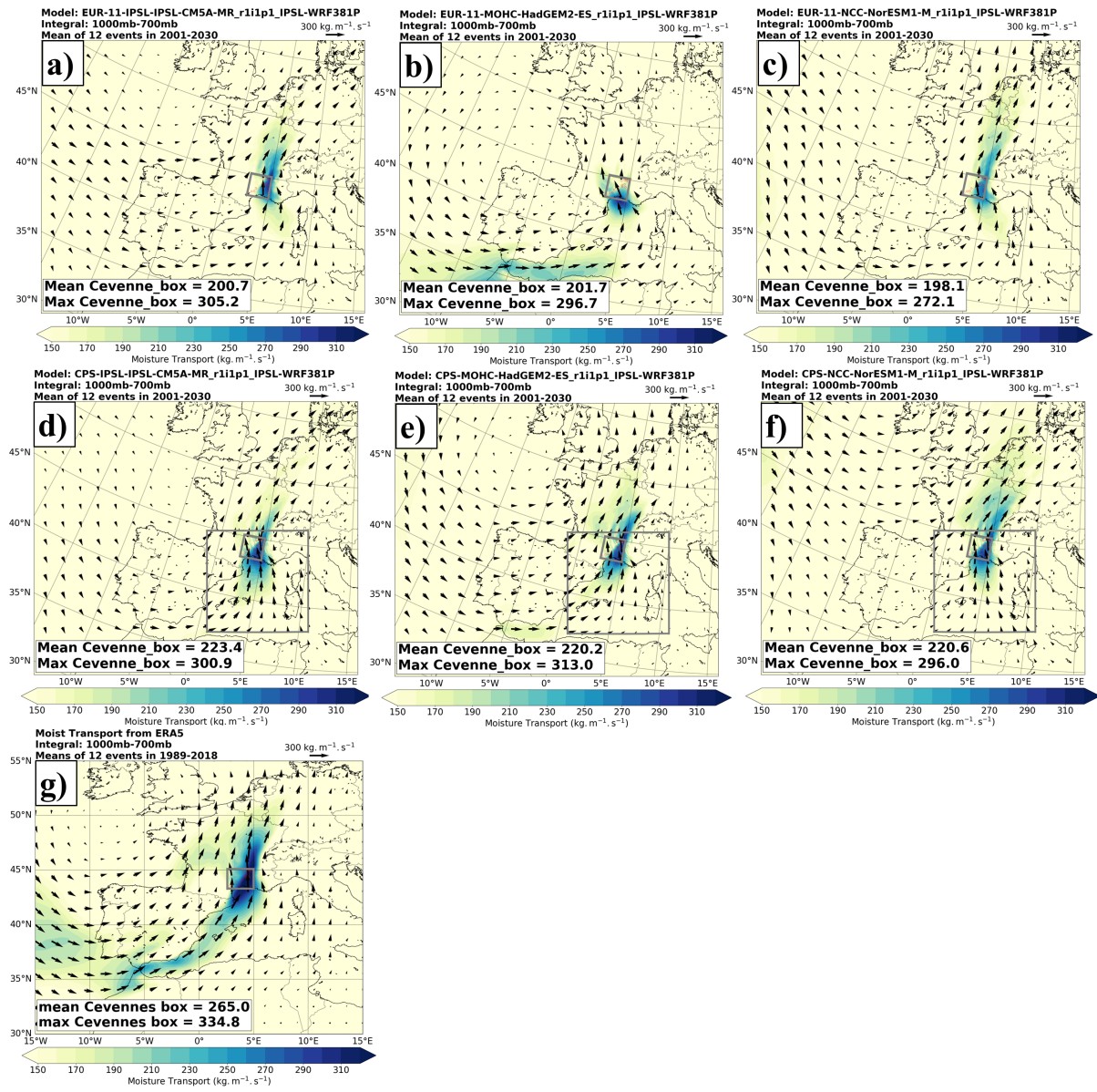

**Figure 7.** Mean moisture transport of the 12 heaviest daily rainfall events from all simulations (2001-2030, from a to c for EUR-11 simulations, and from d to f for CPSs) and ERA5 (1989-2018, panel g). Note that the domain of CPS is far smaller to meet the requirement of these analyses. Therefore, we embedded each CPS moisture transport inside its corresponding driving EUR-11 for the same 12 events of that CPS. This means that results from EUR-11 in these cases (panel d to f) may differ from those from panel a to c.

**Table 1.** The selection of periods of simulations and reference datasets for evaluation of each index.

| No. | Indices | Period for each Dataset | | | | |
|---|---|---|---|---|---|---|
| | | OBS | WRF | COMEPHORE | SAFRAN | ERA5 |
| 1 | Rx1day | 1961-1990 | 1951-1980 | - | 1961-1990 | - |
| 2 | R-T Scaling (daily rainfall) | | | - | - | - |
| 3 | Distribution of wet events (daily rainfall) | | | - | - | - |
| 4 | Rx3hour | 1998-2018 | 2001-2030 | 1997-2007 | - | - |
| 5 | R-T Scaling (daily maximum 3-hour rainfall) | | | - | - | - |
| 6 | Distribution of wet events (daily maximum 3-hour rainfall) | | | - | - | - |
| 7 | Moisture source | 1989-2018 | 2001-2030 | - | - | 1989-2018 |