# Peer review of "Evaluation of convection-permitting extreme precipitation simulations for the south of France"

_Earth System Dynamics, 2020_

## Referee Comment (RC1) · Anonymous Referee #1 · 1 Dec 2020

General comments: The authors use four indices to evaluate the skills of convection-permitting models and EURO-CORDEX in reproducing daily and sub-daily heavy precipitation over the Mediterranean region. As expected, the models with higher resolution which are able to resolve deep convection show better performance. The results are meaningful, and the paper is well written.

Major comments: 1.The evaluation between the simulation and the observation do not cover the same period. On Line 85-90, the authors mentioned "Each convection-permitting simulation ( hereafter mentioned as CPS ) is conducted for two different periods including 1951-1980 and 2001 -2030 with the RCP8.5 scenario for the year

after 2005. These two periods are chosen with a gap period ( 1981-2000) rather than a seamless one in order to perform a climate change impact study which will be presented in another article." Since the climate change impact is not studied in this paper, why do the authors select 2000-2030 simulation to compare with the observation in 1997-2007 (Figure 2,6)? If the period is not the same, are the quantitative results in the paper robust? And why RCP8.5? How much difference between the RCP8.5 and RCP4.5?

2.Besides quantitative evaluation, could the authors add more discussion that could explain the results, tie the results into the scientific literature and emphasize the importance of the results?

Minor comments:

1.Line 55. Could the authors give more specific introduction about the region? Why do the authors select this region to study?

2.Line 92, What does the "Mediterranean events" mean?

3.Line 183, could the authors mark the French Alps in Figure 2? "The EUR-11-HadGem2-ES or CPS-HadGEM2-ES show the best agreement with observations." Did the authors mean the results of French Alps? Could the authors provide quantitative evidence? Like spatial correlation?

4.Section 3.1, could the authors give some explanation about why the EUR-11 performs better than the CPS which resolve better deep convection in French Alps?

5.I think it might be better if the authors exchange the order of 3.3 and 3.4. In the method parts, the second indice is comparing the distribution of wet events.

6.Line 244. Could the authors give more explanation about "the convection scheme used in EUR-11 over-simplified the cloud process".

---

## Referee Comment (RC2) · Anonymous Referee #2 · 29 Jan 2021

This manuscript describes a number of long-term simulations with a convective-permitting model over southern France focussing on its ability to reproduce extreme precipitation events in fall. The study, in which several model runs covering a large number of years with forcing data from different global climate models, is clearly interesting and results indicate that this model is more suited to simulate such events compared to standard high-resolution RCMs. As such studies are sparse this one can be an important contribution that merits publication.

However, the manuscript first needs some improvement. This is partly related to the language, I would suggest a thorough language check before re-submission. It can

also be clearer what has been done (exactly which model version has been run?) and why (the choice of time periods including the mismatch with the observations used for evaluation). It is also not clearly explained how this high-resolution convection-permitting model performs when driven by perfect boundary conditions. As it is now, it is not clear if biases are related to poor forcing conditions (including wind, stability, SSTs) from the GCMs or if they result from poor model performance. The same is true for the coarser-scale EUR-11. The results in some of the figures are not entirely in line with how the text describes them.

As for the structure of the paper, I find that there is no proper discussion of the results. Currently, there are some references alluded to and compared with both in the result section and in the conclusion chapter. I think that the discussion should go into either the results section or be introduced in a separate chapter of its own. Furthermore, the supplementary material is interesting and I'm thinking that it may be useful to include directly in the paper instead (it could be part of the discussion), the paper is not that extensive in its present form.

Specific comments:

Line 22 Please explain what is meant by "cloud-resolving". Most convective clouds are smaller than 3x3 km and are definitely not resolved by the convective-permitting models used here.

Line 31-37 Now also shown for higher latitudes in Scandinavia (Lind et al., 2020, see https://link.springer.com/article/10.1007/s00382-020-05359-3)

Line 42 I would not use "large" and "robust" here to describe the number of simulations done and the status of knowledge. The number is in fact highly limited and only for a few regions mainly covering parts of the mid-latitudes.

Line 56-57 Instead of referring to a project (HyMex) I think it is more interesting for a reader to learn stg on what scientific questions are being addressed and/or why this

is interesting from a societal perspective (the references given may be good here but I don't see the need for introducing the project).

Line 87-88 This is unclear. Why is the model run for 1951-1980 and not 1961-1990? Later results are compared to observations from 1961-1990 and even if the results are from GCM-driven simulations there are forcing differences between these periods potentially compromising the comparison. This should be addressed in the paper. Furthermore, the use of 2001-2030, that is mostly based on a future scenario (RCP8.5) is also not clearly explained and in a way difficult to understand. In a similar way, comparison is done between observations covering 1998-2017 and model simulations covering 2001-2030. Again, there is a mismatch of more than 10 years in a period with a strong global mean change. Is there any implication for the results from this (mismatch in extremes as simulated in the 2020ies with stronger forcing compared to the previous 20 years)?

Line 91 Here, Cévennes is mentioned for the first time. For the not-so-very-French reader it is not clear where these mountains are. It becomes clearer when looking at subsequent figures. But, it would be good already in Figure 1 to illustrate where these mountains are (as part of the Central Massif – I guess?). Also the "Cevennes-box" could be given there. (Reference could also be given to this figure on line 141 where the box is detailed in the text).

Line 100 What is "the French Mediterranean Sea"? Is this stg outside of the territorial 12 nm zone?

Line 105-109 It is not clearly described in the paper how the current configuration of WRF 3.8.1 performs w.r.t. the observed precipitation extremes in any ERA-Interim driven simulation. The supplementary material holds such ERA-Interim driven simulations, however, it should be better addressed at some point in the paper how this model (and the currently used setup) works. Also, why are these particular schemes mentioned here and not others? Is it clear from reading these few lines exactly which

version of the model that has been used? Could someone else reproduce your experiment based on what is written here? On line 106 it says SSTs are updated every 6 hours. Is this also true for the lateral boundary conditions?

Figure 1 Why is there no altitude associated with Menorca and Ibiza on the map. Are these islands not resolved by the model?

Line 146 Unclear what "few" means here. Is it only a few time steps from the 6480 time steps (27 hours times 60 minutes times 4 time steps per minute)? Or do you mean that the 6480 time steps are few relative to the full length of the simulations?

Line 157 A reference is missing for ERA5.

Line 227-240 I don't fully agree on the interpretation of the figures including the temperature intervals given in the text here. For instance, in Fig 4a I think it is quite clear that the approximate CC-scaling holds between 3-13C. Between 13 and 18 there seems to be no such relation, but rather a constant precipitation rate regardless of temperature. Similarly, for 4b I think the super-CC scaling applies up to approximately 13C whereafter CC-scaling applies. Furthermore, it is not clear that the models reproduce the behaviour, in some aspects yes but not in the details. The slopes do differ. Also, the slopes differ between EUR-11 and CPS model versions (e.g. HadGEM). This text needs revision. I also think it would be easier to follow if the figure was remade so that the corresponding EUR-11 and CPS simulations (driven by the same GCM) where colored in the same way (suggested to be denoted with full and dashed lines).

Line 243-244 Here it says that "We could explain ... underestimation by the fact ... simplified cloud process". I don't see how this is explained here! Please be more explicit.

Line 262 This section about moisture sources would be a good place to say stg more explicit about the underlying GCMs. From the figure it appears that the Hadley model have a better representation of the moisture flux over the southern parts of

the Mediterranean in association to the events examined. Another feature that could be addressed would be SSTs of the GCMs in association with the events. If some of them have strong biases it would likely influence the moisture source and transport. The moisture supply from the sea is of course very important in this aspect (as also shown in a convective-permitting model for this area by Lenderink et al., 2019, https://iopscience.iop.org/article/10.1088/1748-9326/ab214a/pdf)

Detailed minor comments:

In general, the manuscript needs a careful revision of the language. There are many examples of errors and/or things that could be clarified, some of which are given below.

Line 8 Change "downscaled" to "run"

Line 10 Remove the first "simulations"

Line 24 Change into "is also hope"

Line 25 Consider changing to "conducting several runs to generate large ensembles of simulations with sufficient resolution"

Line 45 Instead of "surface field" I would suggest "surface properties"

Line 49 Remove "in the simulation results"

Line 59 Here is an example of a language problem where it says "Z et al found that convective-permitting model outperformed . . .". Either it should be "a convection-permitting model" or "convection-permitting models".

Line 72-73 This is difficult to understand. The "analysis" is not "downscaling results of EURO-CORDEX" as it says. Rather the analysis is undertaken on results from downscaling EURO-CORDEX simulations.

Line 77-78 This is not really needed in this short paper that is quite standard in its structure. In case it is retained it could be explicitly mentioned that there is also supporting

material and what can be found there (and why not in the paper itself?)

Line 84-85 Shorter with ". . .EURO-11 simulations were also done with WRF-ARW version 3.8.1 driven by three general circulation models (GCMs). . ."

Line 128 It is not the "moisture sources" but the "moisture" that is transported from the Med Sea. And on the same line not the "massive moisture" but the "massive amount of moisture".

Line 136 Suggest to replace "zonal and meridional" with "horizontal"

Line 137 "hPa" instead of "mb"

Line 251-252 I think the () can be removed here. References to the figures are given appropriately in the subsequent text.

Line 255 Should it be "-45%" here instead of 40?

Line 263 It is not the ability of the "simulation" but of the "model" that is investigated.

---

## Referee Comment (RC3) · Anonymous Referee #3 · 2 Feb 2021

**Review of "Evaluation of convection-permitting extreme precipitation simulations for the south of France" by Luu et al. (2021)**

The authors perform an evaluation study of a convection-permitting model (CPM) at 3 km resolution. The simulation domain covers the south-east of France and part of the Mediterranean Sea. The CPM downscales a 0.11° model, which was run over the EURO-CORDEX domain. A nice aspect of the study is that there are three realisations of the CPM simulations, each based on a different GCM; this aspect could potentially be given mode attention, as it is unusual in the literature. The authors then evaluate the performance of the CPM and 0.11° models and conclude that the CPM produces more realistic precipitation.

I think the study is a reasonable contribution to the literature and could in principal be published. However, at present the study has some limitations which must first be addressed before publication. These are detailed below in the main comments section.

**Main Comments**

**1. Novelty and relation to similar literature.** In their abstract, the authors state of their climate-length convection-permitting simulations that "… *this approach has never been used in a climate simulation for the Mediterranean coastal region*" (L4-5). There's a similar statement in the Introduction (L69-70): "… *such long simulations, to the best of our knowledge, have never been done for coastal area in the Mediterranean region*".

This is not correct. I can think of at least five studies which perform convection-permitting simulations at climate timescales over the north-western Mediterranean, which the authors don't cite. These studies all cover the area of the CPM domain used by the authors, as opposed to the studies of e.g. Armon et al. (2020) and Zittis et al. (2017) cited by the authors, which are for other parts of the Mediterranean and aren't on climate timescales. The studies I have in mind are (there may be more):

[1] Berthou et al.: https://doi.org/10.1007/s00382-018-4114-6
[2] Vergara-Temprado et al.: https://doi.org/10.1029/2020GL089506
[3] Meredith et al.: https://doi.org/10.1088/1748-9326/ab6787
[4] Adinolfi et al.: https://doi.org/10.3390/atmos12010054
[5] Caillaud et al.: https://doi.org/10.1007/s00382-020-05558-y

Ref. [1] has a specific section on heavy precipitation events in SE France. Refs. [4] and [5] also assess intense hourly and daily precipitation events in CPMs over France using similar observation sets to the present authors. Ref. [3] uses the same annual re-initialization technique as the authors and also focuses on the Autumn months in the NW Mediterranean, just as the present authors do.

Around lines 55-67 it would also be good to cite these climate-scale studies, as most of those presently cited are for case studies or selected events.

The authors need to cite and discuss the relevant literature, not just in the Introduction, but also where appropriate in the Results and Discussion. The results of the present authors should be presented in the context of the pre-existing relevant literature. That means, wherever appropriate, compare your results with those in the pre-existing literature. This is particularly important if your results are different, in which case possible explanations would be helpful.

**2. Comparison of model data and observations at different spatial scales.**

A major issue with the evaluation is that model data and observations on different spatial scales are being compared directly. While it's arguable that model data on a 3 km grid could be compared directly with station data, what do the authors hope to learn by comparing data on a 12 km grid (that means grid box averages over an area of 12 x 12 = 144 km$^2$) with station data (point values)? Or even with the 1 km COMEPHORE product?

It is not surprising that Figures 4, 5 and 6 show the lowest intensities in the 12 km model, followed by the 3 km model, followed by the point observations. This simply reflects the fact that the extremes are being averaged over ever greater areas as the grid spacing increases, thus the intensities are "smoothed out"; the same applies to the "mean 14/23 stations" in Figs. 2 and 3. Indeed this also applies to the box means in Figs. 2 and 3, because the area mean of high-resolution extremes must by higher than the area-mean of low-resolution extremes. These comparisons don't tell us whether or not the 12 km model is worse than the 3 km model, or vice versa. Suppose your 12 km model was perfect at the 12 km scale: the extreme intensities would still be much lower than those at the 3 km or point scale. Or imagine you aggregated your 1 km COMEPHORE data to the 12 km grid and then compared it against the 1 km data at some point: the 12 km data would have a strong negative bias, even though it's the same dataset. For further discussion of this topic, I suggest the study of Göber et al. (2008, https://doi.org/10.1002/met.78).

In the case of Fig. 4 (temperature scaling), what's important is that the models have similar scaling curves to observations, the intensities don't need to match to validate the models.

As pointed out in Göber et al. (2008), the standard/appropriate way to compare observations and model data is by upscaling the observations to the coarsest model grid (EUR-11 in your case). The CPS publications the authors cite all upscale their observations to the coarsest model grid: Kendon et al. (2012), Fosser et al. (2015), Knist et al. (2018), Chan et al. (2013, 2014). Also Refs. [1], [4] and [5] above.

In the cases of the gridded observations (SAFRAN, COMEPHORE), it is certainly possible to compare models and observations at the same spatial scale (i.e. that of EUR-11) through conservative remapping. In the case of the station data, there's no simple solution. As stated above, comparing the 3 km intensities with stations could be defensible. I don't see much value in comparing the 12 km intensities with stations; but if the authors really want to do this then they need to give a very strong warning to the reader that this has limitations, and these limitations should be communicated in the text.

Another indicator that the results might be being affected by the comparison of different spatial scales is the added value you find for daily precipitation. Studies show that CPMs generally don't add value for daily mean or extreme precipitation, e.g. Refs. [1] and [4] above, Chan et al. (2013), Ban et al. (2014). It's likely that a lot of the added value you find for daily precipitation statistics is simply due to the different spatial scales you're comparing against observations. Having said that, Berthou et al. (2018, Ref. [1]) did find added value at the daily scale for CPMs in the case of autumnal precipitation extremes in the Mediterranean.

**Other Comments**

1. Ideally this study would have been performed using reanalysis as boundary forcing. Since you are using free-running GCMs, you therefore need to inform the reader early on (i.e. in the methods) that the regional models will inherit biases from the GCMs, and that any biases you find therefore

result from a combination of both the GCM and RCM biases. Later on in your results, we see quite different results depending on what the GCM is, so the role of the GCM is clearly not trivial.

2. The CPM simulations cover the Autumn months because this is the time when the most intense events occur in SE France. Maybe not all readers will be aware of this or know why, as many expect the most intense short-duration events to be in the summer. I think a few sentences in the Introduction and/or Methods explaining why the strongest events are in Autumn would be useful. E.g. warmer Mediterranean SSTs, low pressure systems advecting warm moist air at lower levels from the Mediterranean into southern France and then orographic lifting, etc. Maybe the studies of Labeaupin et al. (2006, https://doi.org/10.1029/2005JD006541) and Toreti et al. (2010, https://doi.org/10.5194/nhess-10-1037-2010) would be of interest to you.

3. Temperature scaling of extreme precipitation (L118-126). What steps have you taken in order to avoid effects from under-sampling? Do you require some minimum value of data points to be in a bin before you compute the percentile? If so, what? Boessenkool et al. (2017, https://doi.org/10.5194/nhess-17-1623-2017) show that the downturn at higher temperatures can simply be a statistical artefact if the bins are not sufficiently populated. In your Figure 4, the deviations away from CC or 2xCC scaling occur at low and high temperatures, exactly the range where there are less events. This could be due to insufficient data points in the bins.

Also, in Figure 4, do the numbers in the inset table represent the mean scaling rates? If so, how do you compute them? Over the entire range of data? Or is it an average across all stations?

4. There are lots of different data sets used: Gridded data, 14 stations, 23 stations, etc. When the biases are presented in the text (Section 3.2), it is sometimes not clear with respect to which data the bias is for. It might help the reader if you state this more explicitly in the text.

**Minor Comments**

L15-16: *"because of the limitation in computer resources, deep convection processes have rarely been solved explicitly in long climate simulations"*. This is again a bit of an exaggeration with respect to the existing literature. There are really quite a lot of CPM studies on climate timescales. For example, there are the studies which you already cite: Ban et al. (2014, 2015), Fosser et al. (2015), Hodnebrog et al. (2019), Kendon et al. (2014), Knist et al. (2018), Vanden Broucke et al. (2019). Then there are the five I've listed under Main Comment 1. There are a lot more if you take a look on Google Scholar, and not just for Europe like those already listed.

L28-44: Please remember to also cite literature relevant to your study region.

L41: "added value" is always singular, i.e. not "added values". Also in other parts of the manuscript.

L82: Could you please also give the resolution of the CPM in degrees?

L103: Could you please state what the model top is? With only 32 levels, the spacing between layers could be quite high. You should avoid having a vertical spacing which is greater than your horizontal spacing, which may be a risk here for your CPM simulations. It's too late to change this now, but it's useful to keep in mind for the future.

L105-109: Is the shallow convection parametrized in the CPM? If so, what scheme?

L124: Maybe you mean "same" instead of "similar"? "Similar" doesn't mean "identical", but "same" does.

L136: Unit of g is m s$^{-2}$.

L148-163: The authors could consider making these lines into a separate Section 2.3 for the data sets? If they don't want to, that's also OK.

L178-180: Do these biases refer to the bias over the whole box against SAFRAN? If so, the numbers don't agree with my calculations based on the insets in the panels of Fig. 2. please check.

L203-205: Are these 23 stations for the time period in 3 (h) or 3(j)?

L220: Instead of "we model the Clausius-Clapeyron relation …", it would be more correct to say "we investigate if the temperature-precipitation scaling follows the Clausius-Clapeyron relation in observations and models", or similar.

L230: The EUR-11 model can't be expected to have similar intensities as the point-scale observations, simply because you're comparing at different scales here (see main comment 2). What's important is whether the EUR-11 and CPM have the same scaling rate. Same goes for L243.

L235-240: Maybe your super-CC scaling results from the combination of strong moisture convergence in autumn precipitation extremes in SE France (due to onshore moisture advection) and deep convection. These ingredients aren't present simultaneously at other times of the year.

-L249 (Section 3.4): My understanding is that the analysis in this section is based on wet-events, i.e. days without precipitation are excluded. If this is the case, it would be useful for the reader to know what fraction of days contain wet events and if this differs much between the different simulations.

-L252: Change "either … or" to "both … and".

-Figure 3: There's no panel (i) after (h), so I think you need to change (j) to (i).

-Figure 3: What does the yellow colour over Italy represent? If this is simply an area of no data, then it would be good to mask it in white like in Figure 2 (g).

---

## Author Comment (AC1) · 7 Jun 2021

General comments: The authors use four indices to evaluate the skills of convectionpermitting models and EURO-CORDEX in reproducing daily and sub-daily heavy precipitation over the Mediterranean region. As expected, the models with higher resolution which are able to resolve deep convection show better performance. The results are meaningful, and the paper is well written.

Response: We thank the reviewer for the appreciation of our work.

Major comments:

1. The evaluation between the simulation and the observation do not cover the same period. On Line 85-90, the authors mentioned "Each convection- permitting simulation (hereafter mentioned as CPS) is conducted for two different periods including 1951-1980 and 2001-2030 with the RCP8.5 scenario for the year after 2005. These two periods are chosen with a gap period (1981-2000) rather than a seamless one in order to perform a climate change impact study which will be pre- sented in another article." Since the climate change impact is not studied in this paper, why do the authors select 2000-2030 simulation to compare with the observation in 1997-2007 (Figure 2,6)? If the period is not the same, are the quantitative results in the paper robust? And why RCP8.5? How much difference between the RCP8.5 and RCP4.5?

Response: 1) As we stated in Table 1 of the preprint, we had two separated periods for in situ observations. First, for the daily timescale dataset, most stations started in 1961 and spanned to 2014. This dataset was adopted from Vautard et al. 2015. Second, the daily maximum of 3-hourly rainfall dataset (daily value of maximum 3-hour time window of rainfall), which was collected lately, started almost in 1998 to 2018. Therefore, we evaluated the daily indices of historical simulations (1951-1980) against observations of 1961-1990 and evaluated the 3-hourly indices of current period simulations against observations of 1998-2018. Because those simulations were forced by CMIP5 models and then evaluated by the mean state of the periods, the slight difference (5 to 10 years) in periods among models and observations does not hinder comparison. Thus, the quantitative results in this study are robust. 2) Chapter 12 in IPCC-AR5 (Collins et al., 2013) showed that anthropogenic radiative forcing started to diverge only after 2030 that also led to the divergence of global mean surface temperature after this year. Therefore, the discrepancy of using the CMIP5 simulations under different RCPs to force regional climate models for the period before 2030 is trivial. We then can consider them as different realizations of weather for a specific climate state.

2. Besides quantitative evaluation, could the authors add more discussion that could
explain the results, tie the results into the scientific literature and emphasize the importance of the results?

Response: We will add further discussion to section 3 in our revised manuscript.

Minor comments:

1. Line 55. Could the authors give more specific introduction about the region? Why do the authors select this region to study?

Response: We would replace a sentence starting in line 55 by 3 sentences providing the motivation why the Mediterraen region has been receiving more interest and specific scientific questions are being addressed by research communities.

"The coastal regions along the Mediterranean frequently undergo heavy precipitation events in the autumn which subsequently lead to flash floods and landslides causing massive losses and damages (Delrieu et al., 2005; Fresnay et al., 2012; Llasat et al., 2013; Nuissier et al., 2008; Ricard et al., 2012). In addition, this area is considered as a hotspot of climate change that strongly responds to warming at global scale (Giorgi, 2006; Tuel and Eltahir, 2020). As a result, the Mediterranean has received an increasing scientific interest in understanding of mechanisms leading to flood-inducing heavy precipitation as well as in improving the model ability to predict and project those events in a complex changing climate that provides substantial support to adaptation and mitigation for society (Drobinski et al., 2014; Ducrocq et al., 2014)."

2. Line 92, What does the "Mediterranean events" mean?

Response: The "Mediterranean events" here denote extreme precipitation events in the Mediterranean coastal areas. We have clarified this in the main text of the article.

3. Line 183, could the authors mark the French Alps in Figure 2? "The EUR-11-HadGem2-ES or CPS-HadGEM2-ES show the best agreement with observations." Did the authors mean the results of French Alps? Could the authors provide quantitative evidence? Like spatial correlation? Interactive comment

Response: The French Alps is located at the boundary of France, Italy and Switzerland, and noticeable in Fig.1 of this response and also the Figure 1 in the preprint. However, We meant that those two simulations show the best agreement with observations over the Cévennes box when comparing them with other simulations with the same resolution (Fig.1). The two downscaling experiments from HadGEM2-ES show dry biases of -5.9% for CPS and -33% for EUR-11 over the Cévennes box whose absolute values are smallest compared to others in the same resolution.

4. Section 3.1, could the authors give some explanation about why the EUR-11 performs better than the CPS which resolve better deep convection in French Alps?

Response: Biases concerned in the french Alps are difficult to interpret due to (1) the large heterogeneity of terrain and presence of high mountains in the area of the trends, and (2) the yet coarse resolution of models (even with CPS configuration) relative to mountains.

5. I think it might be better if the authors exchange the order of 3.3 and 3.4. In the method parts, the second indice is comparing the distribution of wet events.

Response: We will switch the position of section 3.3 and 3.4 in the main text.

6. Line 244. Could the authors give more explanation about "the convection scheme used in EUR-11 over-simplified the cloud process".

Response: The complexity of updraft in mesoscale convective systems was described in Houze (2004). However, the convection schemes usually simplify and formulate these complex processes by statistical distributions. These schemes use information from large-scale variables from model grids to modulate the development of convective cells at a finer scale that cannot be resolved by model resolution (Westra et al., 2014). This also implies assumptions of quasi-equilibrium with large-scale forcing, approximation of moist air entraining in the updraft, and representation of all single cloud elements by sole steady state updraft of the whole cloud ensemble (Houze, 2004; Lenderink and
Attema, 2015; Prein et al., 2013; de Rooy et al., 2013). In addition, convection schemes can respond to instant changes in atmospheric instability through information from grid scale, however, they do not memorize the previous state. This leads to their inability to permit the advection, development or decay of convective storms (Westra et al., 2014). An overview of historical development of assumption/parameterization of convection schemes was presented in de Rooy et al., (2013).

Please find Fig.1 at the end of this document.

Reference

Delrieu, G., Nicol, J., Yates, E., Kirstetter, P.-E., Creutin, J.-D., Anquetin, S., Obled, C., Saulnier, G.-M., Ducrocq, V., Gaume, E., Payrastre, O., Andrieu, H., Ayral, P.-A., Bouvier, C., Neppel, L., Livet, M., Lang, M., du-Châtelet, J. P., Walpersdorf, A. and Wobrock, W.: The Catastrophic Flash-Flood Event of 8–9 September 2002 in the Gard Region, France: A First Case Study for the Cévennes–Vivarais Mediterranean Hydrometeorological Observatory, J. Hydrometeorol., 6(1), 34–52, doi:10.1175/jhm-400.1, 2005.

Drobinski, P., Ducrocq, V., Alpert, P., Anagnostou, E., Béranger, K., Borga, M., Braud, I., Chanzy, A., Davolio, S., Delrieu, G., Estournel, C., Boubrahmi, N. F., Font, J., Grubišić, V., Gualdi, S., Homar, V., Ivančan-Picek, B., Kottmeier, C., Kotroni, V., Lagouvardos, K., Lionello, P., Llasat, M. C., Ludwig, W., Lutoff, C., Mariotti, A., Richard, E., Romero, R., Rotunno, R., Roussot, O., Ruin, I., Somot, S., Taupier-Letage, I., Tintore, J., Uijlenhoet, R. and Wernli, H.: HyMeX: A 10-Year Multidisciplinary Program on the Mediterranean Water Cycle, Bull. Am. Meteorol. Soc., 95(7), 1063–1082, doi:10.1175/bams-d-12-00242.1, 2014.

Ducrocq, V., Braud, I., Davolio, S., Ferretti, R., Flamant, C., Jansa, A., Kalthoff, N., Richard, E., Taupier-Letage, I., Ayral, P.-A., Belamari, S., Berne, A., Borga, M., Boudevillain, B., Bock, O., Boichard, J.-L., Bouin, M.-N., Bousquet, O., Bouvier, C., Chiggiato, J., Cimini, D., Corsmeier, U., Coppola, L., Cocquerez, P., Defer, E., Delanoë,

**ESDD**
J., Girolamo, P. Di, Doerenbecher, A., Drobinski, P., Dufournet, Y., Fourrié, N., Gourley, J. J., Labatut, L., Lambert, D., Coz, J. Le, Marzano, F. S., Molinié, G., Montani, A., Nord, G., Nuret, M., Ramage, K., Rison, W., Roussot, O., Said, F., Schwarzenboeck, A., Testor, P., Baelen, J. Van, Vincendon, B., Aran, M. and Tamayo, J.: HyMeX-SOP1: The Field Campaign Dedicated to Heavy Precipitation and Flash Flooding in the Northwestern Mediterranean, Bull. Am. Meteorol. Soc., 95(7), 1083–1100, doi:10.1175/bams-d-12-00244.1, 2014.

Fresnay, S., Hally, A., Garnaud, C., Richard, E. and Lambert, D.: Heavy precipitation events in the Mediterranean: sensitivity to cloud physics parameterisation uncertainties, Nat. Hazards Earth Syst. Sci., 12(8), 2012.

Giorgi, F.: Climate change hot-spots, Geophys. Res. Lett., 33(8), doi:10.1029/2006gl025734, 2006.

Houze, R. A.: Mesoscale convective systems, Rev. Geophys., 42(4), 1–43, doi:10.1029/2004RG000150, 2004.

Lenderink, G. and Attema, J.: A simple scaling approach to produce climate scenarios of local precipitation extremes for the Netherlands, Environ. Res. Lett., 10(8), 85001, 2015. Llasat, M. C., Llasat-Botija, M., Petrucci, O., Pasqua, A. A., Rosselló, J., Vinet, F. and Boissier, L.: Towards a database on societal impact of Mediterranean floods within the framework of the HYMEX project, Nat. Hazards Earth Syst. Sci., 13(5), 1337–1350, doi:10.5194/nhess-13-1337-2013, 2013.

Nuissier, O., Ducrocq, V., Ricard, D., Lebeaupin, C. and Anquetin, S.: A numerical study of three catastrophic precipitating events over southern France. I: Numerical framework and synoptic ingredients, Q. J. R. Meteorol. Soc., 134(630), 111–130, 2008. Prein, Gobiet, A., Suklitsch, M., Truhetz, H., Awan, N. K., Keuler, K. and Georgievski, G.: Added value of convection permitting seasonal simulations, Clim. Dyn., 41(9–10), 2655–2677, 2013.
Ricard, D., Ducrocq, V. and Auger, L.: A Climatology of the Mesoscale Environment Associated with Heavily Precipitating Events over a Northwestern Mediterranean Area, J. Appl. Meteorol. Climatol., 51(3), 468–488, doi:10.1175/JAMC-D-11-017.1, 2012. de Rooy, W. C., Bechtold, P., Fröhlich, K., Hohenegger, C., Jonker, H., Mironov, D., Pier Siebesma, A., Teixeira, J. and Yano, J.-I.: Entrainment and detrainment in cumulus convection: an overview, Q. J. R. Meteorol. Soc., 139(670), 1–19, doi:https://doi.org/10.1002/qj.1959, 2013.

Tuel, A. and Eltahir, E. A. B.: Why Is the Mediterranean a Climate Change Hot Spot?, J. Clim., 33(14), 5829–5843, doi:10.1175/JCLI-D-19-0910.1, 2020.

Westra, S., Fowler, H. J., Evans, J. P., Alexander, L. V, Berg, P., Johnson, F., Kendon, E. J., Lenderink, G. and Roberts, N. M.: Future changes to the intensity and frequency of short-duration extreme rainfall, Rev. Geophys., 52(3), 522–555, doi:10.1002/2014rg000464, 2014.

**ESDD**

---

## Author Comment (AC2) · 7 Jun 2021

The authors perform an evaluation study of a convection-permitting model (CPM) at 3 km resolution. The simulation domain covers the south-east of France and part of the Mediterranean Sea. The CPM downscales a 0.11° model, which was run over the EURO-CORDEX domain. A nice aspect of the study is that there are three realisations of the CPM simulations, each based on a different GCM; this aspect could potentially be given mode attention, as it is unusual in the literature. The authors then evaluate the performance of the CPM and 0.11° models and conclude that the CPM produces more realistic precipitation.

I think the study is a reasonable contribution to the literature and could in principal be published. However, at present the study has some limitations which must first be addressed before publication. These are detailed below in the main comments section.

**Response:** We thank the reviewer for all the constructive comments and suggestions which do help us to improve our manuscript.

Main Comments

1. Novelty and relation to similar literature. In their abstract, the authors state of their climate-length convection-permitting simulations that "... this approach has never been used in a climate simulation for the Mediterranean coastal region" (L4-5). There's a similar statement in the Introduction (L69-70): "... such long simulations, to the best of our knowledge, have never been done for coastal area in the Mediterranean region".

This is not correct. I can think of at least five studies which perform convection-permitting simulations at climate timescales over the north-western Mediterranean, which the authors don't cite. These studies all cover the area of the CPM domain used by the authors, as opposed to the studies of e.g. Armon et al. (2020) and Zittis et al. (2017) cited by the authors, which are for other parts of the Mediterranean and aren't on climate timescales. The studies I have in mind are (there may be more):

[1] Berthou et al.: https://doi.org/10.1007/s00382-018-4114-6

[2] Vergara-Temprado et al.: https://doi.org/10.1029/2020GL089506

[3] Meredith et al.: https://doi.org/10.1088/1748-9326/ab6787

[4] Adinolfi et al.: https://doi.org/10.3390/atmos12010054

[5] Caillaud et al.: https://doi.org/10.1007/s00382-020-05558-y

Ref. [1] has a specific section on heavy precipitation events in SE France. Refs. [4] and [5] also assess intense hourly and daily precipitation events in CPMs over France using similar observation sets to the present authors. Ref. [3] uses the same annual re-initialization technique as the authors and also focuses on the Autumn months in the NW Mediterranean, just as the present authors do.

Around lines 55-67 it would also be good to cite these climate-scale studies, as most of those presently cited are for case studies or selected events.

The authors need to cite and discuss the relevant literature, not just in the Introduction, but also where appropriate in the Results and Discussion. The results of the present authors should be presented in the context of the pre-existing relevant literature. That means, wherever appropriate, compare your results with those in the pre-existing literature. This is particularly important if your results are different, in which case possible explanations would be helpful.

**Response:** We thank you for suggesting more new studies investigating convection-permitting simulations, especially their domains that cover the French Mediterranean region. They are helpful for us in improving our introduction as well as provide new material to discuss our results. And we will try to look up more concerning studies to improve our discussion.

We will modify our statement in both the abstract and introduction. For the sentence in line 4 and 5 "However, this approach has never been used ...", we will change to "*This approach has been tested and performed at climate scale in several studies in recent decades for different areas*". We will also remove the sentence in line 69 and 70 "However, such long simulations, ... for coastal areas in the Mediterranean region".

2. Comparison of model data and observations at different spatial scales.

A major issue with the evaluation is that model data and observations on different spatial scales are being compared directly. While it's arguable that model data on a 3 km grid could be compared directly with station data, what do the authors hope to learn by comparing data on a 12 km grid (that means grid box averages over an area of 12 x 12 = 144 km2) with station data (point values)? Or even with the 1 km COMEPHORE product?

**Response:** We have noticed that there are discrepancies in spatial scales among CPSs, EUR-11s and observations datasets. We agree that the spatial averaging effect could smooth out extremes. However, from a model user perspective, one would compare directly what models produce to local observations to serve, e.g. the climate impact at local scale. For instance many users directly use reanalyses with resolutions from 5 to 50 km for local studies, possibly coupled with some statistical techniques. For such applications, the fact that a higher resolution model has a larger spatial variability is to be counted in the added value of the model. By using a higher resolution without a convection parameterization in model configuration, we found here the improvement in reproducing the extreme in smaller spatial and temporal scales. This improvement can in part be explained not only by removing averaging effects, but also other factors coming into play (e.g. moisture convergence, better resolved mountains and flows). Our goal here is to assess the overall change in comparisons to station data.

It is not surprising that Figures 4, 5 and 6 show the lowest intensities in the 12 km model, followed by the 3 km model, followed by the point observations. This simply reflects the fact that the extremes are being averaged over ever greater areas as the grid spacing increases, thus the intensities are "smoothed out"; the same applies to the "mean 14/23 stations" in Figs. 2 and 3. Indeed this also applies to the box means in Figs. 2 and 3, because the area mean of high-resolution extremes must by higher than the area-mean of low-resolution extremes. These comparisons don't tell us whether or not the 12 km model is worse than the 3 km model, or vice versa. Suppose your 12 km model was perfect at the 12 km scale: the extreme intensities would still be much lower than those at the 3 km or point scale. Or imagine you aggregated your 1 km COMEPHORE data to the 12 km grid and then compared it against the 1 km data at some point: the 12 km data would have a strong negative bias, even though it's the same dataset. For further discussion of this topic,

I suggest the study of Göber et al. (2008, https://doi.org/10.1002/met.78).

**Response:** We disagree with the reviewer at this point (however we did additional analyses to check what the reviewer suggests, see below). Our goal here is to assess the overall improvement against observed station data (resolution and other processes) with CP set-up (see above) against previous approaches. Our goal is not to disentangle causes of improvement and processes which would require a much longer study with other types of data at climatic time scale. In our simulations, the resolution is not the only factor changed, also the convection scheme is switched off. In addition, a few other parameterizations concerning turbulence in the gray zone were developed for coarser resolutions that can lead to additional systematic biases when applied in CPS configuration (Prein et al., 2020). All these mean that the CPS configuration also introduces different changes and potential sources of uncertainty. Comparing 12-km resolution and the CPS at the same low resolution would only answer the question of how the finer resolution can improve larger-scale phenomena.

In the case of Fig. 4 (temperature scaling), what's important is that the models have similar scaling curves to observations, the intensities don't need to match to validate the models.

**Response:** We thank you for this suggestion. Indeed, the convection-permitting simulations are able to reproduce similar scaling patterns to observations that should be mentioned in the text.

As pointed out in Göber et al. (2008), the standard/appropriate way to compare observations and model data is by upscaling the observations to the coarsest model grid (EUR-11 in your case). The CPS publications the authors cite all upscale their observations to the coarsest model grid: Kendon et al. (2012), Fosser et al. (2015), Knist et al. (2018), Chan et al. (2013, 2014). Also Refs. [1], [4] and [5] above.

**Response:** In our opinion, there is no "standard" way of evaluating model performance. This depends on which scientific question is being addressed. In our study, we only focus on how/to what extent the CPSs improve extreme precipitation at a local scale rather than the question why a finer resolution improves the final output. Upscaling simulations at a coarser scale would only partly assess the relative CPSs by answering whether newly-resolved processes improve the coarser scale. We are fully conscious and agree that this is an interesting question, but not the one we address here by comparing simulations with station data.

In the cases of the gridded observations (SAFRAN, COMEPHORE), it is certainly possible to compare models and observations at the same spatial scale (i.e. that of EUR-11) through conservative remapping. In the case of the station data, there's no simple solution. As stated above, comparing the 3 km intensities with stations could be defensible. I don't see much value in comparing the 12 km intensities with stations; but if the authors really want to do this then they need to give a very strong warning to the reader that this has limitations, and these limitations should be communicated in the text.

Another indicator that the results might be being affected by the comparison of different spatial scales is the added value you find for daily precipitation. Studies show that CPMs generally don't add value for daily mean or extreme precipitation, e.g. Refs. [1] and [4] above, Chan et al. (2013), Ban et al. (2014). It's likely that a lot of the added value you find for daily precipitation statistics is simply due to the different spatial scales you're comparing against observations. Having said that, Berthou et al. (2018, Ref. [1]) did find added value at the daily scale for CPMs in the case of autumnal precipitation extremes in the Mediterranean.

**Response:** We performed additional graphs showing results from the CPSs upscaled to EUR-11 resolution using the conservative remapping (referred as CPS-11, see Fig.1 at the end of this document). This additional analysis shows that the CPS-11s still have good agreement to both gridded and in situ observations though their resolutions/scales are different. The CPS-11s give similar biases of statistics concerning the mean of the Cévennes box against mean of stations within the box for Rx1day, while their maxima of box/stations deviate from the observed values by roughly -24% to 14%. This means that the CPSs do improve the results of Rx1day in our study after considering the upscaling analysis. This also confirmed what was found in Fumière et al., (2019) with AROME model forced by reanalysis data. Therefore, we will mention this point in our manuscript and state that our conclusion remains the same when we perform the upscaling investigation for the CPSs.

Other Comments

1. Ideally this study would have been performed using reanalysis as boundary forcing. Since you are using free-running GCMs, you therefore need to inform the reader early on (i.e. in the methods) that the regional models will inherit biases from the GCMs, and that any biases you find therefore result from a combination of both the GCM and RCM biases. Later on in your results, we see quite different results depending on what the GCM is, so the role of the GCM is clearly not trivial.

**Response:** We thank you for raising this point. We will discuss this point in the revised manuscript.

2. The CPM simulations cover the Autumn months because this is the time when the most intense events occur in SE France. Maybe not all readers will be aware of this or know why, as many expect the most intense short-duration events to be in the summer. I think a few sentences in the Introduction and/or Methods explaining why the strongest events are in Autumn would be useful. E.g. warmer Mediterranean SSTs, low pressure systems advecting warm moist air at lower levels from the Mediterranean into southern France and then orographic lifting, etc. Maybe the studies of

Labeaupin et al. (2006, https://doi.org/10.1029/2005JD006541) and Toreti et al. (2010, https://doi.org/10.5194/nhess-10-1037-2010) would be of interest to you.

**Response:** We agree and will add the explanation into the method section of the revised manuscript.

3. Temperature scaling of extreme precipitation (L118-126). What steps have you taken in order to avoid effects from under-sampling? Do you require some minimum value of data points to be in a bin before you compute the percentile? If so, what? Boessenkool et al. (2017, https://doi.org/10.5194/nhess-17-1623-2017) show that the downturn at higher temperatures can simply be a statistical artefact if the bins are not sufficiently populated. In your Figure 4, the deviations away from CC or 2xCC scaling occur at low and high temperatures, exactly the range where there are less events. This could be due to insufficient data points in the bins.

**Response:** We have not applied any rule to avoid under-sampling effect. Therefore, in this revision, we chose a threshold of at least 300 points to take a bin into consideration. This helps to eliminate a few bins in lowest and highest temperature ranges. However, the hook shape remains at high temperature ranges that suggests that the lack of moisture plays a role (Hardwick Jones et al., 2010).

Also, in Figure 4, do the numbers in the inset table represent the mean scaling rates? If so, how do you compute them? Over the entire range of data? Or is it an average across all stations?

**Response:** We pooled all stations together and applied the scaling procedure to obtain what was shown in Figure 4. The numbers in legend show mean scaling rates over all bins for each dataset.

4. There are lots of different data sets used: Gridded data, 14 stations, 23 stations, etc. When the biases are presented in the text (Section 3.2), it is sometimes not clear with respect to which data the bias is for. It might help the reader if you state this more explicitly in the text.

**Response:** All the biases presented in the text came from the comparison of simulations against in situ observations. For daily indices, we compared the simulations with the 14 stations within the Cévennes, while we used 23 stations set for evaluating the 3-hour indices from simulations. We will clarify explicitly in the text.

Minor Comments

L15-16: "because of the limitation in computer resources, deep convection processes have rarely been solved explicitly in long climate simulations". This is again a bit of an exaggeration with respect to the existing literature. There are really quite a lot of CPM studies on climate timescales. For example, there are the studies which you already cite: Ban et al. (2014, 2015), Fosser et al. (2015), Hodnebrog et al. (2019), Kendon et al. (2014), Knist et al. (2018), Vanden Broucke et al. (2019). Then there are the five I've listed under Main Comment 1. There are a lot more if you take a look on Google Scholar, and not just for Europe like those already listed.

**Response:** We will change two sentences from line 15 to 18: "However, because of limitation in computer resources, … with prognostic variables have been designed to represent this process at local scale (Kendon et al., 2012)" into "*However, deep convection processes have been parameterized in simulations at climate scale for a long period of time. The parameterization methods that are based on statistical properties of convection processes within a grid box and their interactions with prognostic variables have been designed to represent this process at local scale (Kendon et al., 2012)."*

L28-44: Please remember to also cite literature relevant to your study region.

**Response:** We will improve that paragraph with literature relevant to the Mediterranean area.

L41: "added value" is always singular, i.e. not "added values". Also in other parts of the manuscript. L82: Could you please also give the resolution of the CPM in degrees?

**Response:** We have replaced "*added values*" by "**added value**" in line 41. The resolution of CPM is 0.0275°. We have added it to the text in line 82.

L103: Could you please state what the model top is? With only 32 levels, the spacing between layers could be quite high. You should avoid having a vertical spacing which is greater than your horizontal spacing, which may be a risk here for your CPM simulations. It's too late to change this now, but it's useful to keep in mind for the future.

**Response:** We thank you for your suggestion. The top model level is 50 hpa, roughly 20 km. We bear in mind that for such a high horizontal resolution, e.g., 3 km in our study, we should raise the number of vertical levels into at least 60.

L105-109: Is the shallow convection parametrized in the CPM? If so, what scheme?

**Response:** We do not use an independent shallow convection scheme in our convection-permitting simulations. We used a similar set of physical schemes, except that deep convection was switched off, to EURO-CORDEX simulations, in which shallow convection was tied to deep convection parameterization. This enables us to assess the added value of explicitly resolving deep convection. However, we keep in mind that the horizontal resolution is insufficient to resolve shallow convection explicitly, and that this issue deserves separate investigations.

L124: Maybe you mean "same" instead of "similar"? "Similar" doesn't mean "identical", but "same" does.

**Response:** We agree and will replace "similar" by "same" in line 124.

L136: Unit of g is m s-2.

**Response:** We thank you for pointing out this error.

L148-163: The authors could consider making these lines into a separate Section 2.3 for the data sets? If they don't want to, that's also OK.

**Response:** It is reasonable to dedicate a separated section describing the data. We will make changed to it.

L178-180: Do these biases refer to the bias over the whole box against SAFRAN? If so, the numbers don't agree with my calculations based on the insets in the panels of Fig. 2. please check.

**Response:** Those numbers refer to the bias of simulations against all stations within the Cévennes box from in situ observations, not SAFRAN.

L203-205: Are these 23 stations for the time period in 3 (h) or 3(j)?

**Response:** We compare simulations with 23 stations from Figure 3-j. Since the results from 2 different periods of COMEPHORE are quite similar, we will remove the one with a shorter period to avoid confusion.

L220: Instead of "we model the Clausius-Clapeyron relation ...", it would be more correct to say "we investigate if the temperature-precipitation scaling follows the Clausius-Clapeyron relation in observations and models", or similar.

**Response:** We will change the sentence in line 220-221 to "*In this section, we model the relation between extreme precipitation and daily mean surface temperature, which is theoretically reflected by the Clausius-Clapeyron equation, by a simple non-parametric scaling method described in section 2.2.*"

L230: The EUR-11 model can't be expected to have similar intensities as the point-scale observations, simply because you're comparing at different scales here (see main comment 2). What's important is whether the EUR-11 and CPM have the same scaling rate. Same goes for L243.

**Response:** The updated scaling analysis shows that the EURO-CORDEX simulations can reproduce scaling rate in case of extreme daily precipitation, while convection-permitting simulations (CPS) tend to slightly overestimate. For extreme sub-daily precipitation, the EURO-CORDEX simulations fail to provide the overall scaling rate, while CPSs show their advantages in capturing convective events.

L235-240: Maybe your super-CC scaling results from the combination of strong moisture convergence in autumn precipitation extremes in SE France (due to onshore moisture advection) and deep convection. These ingredients aren't present simultaneously at other times of the year.

**Response:** In fact, the deep convection in the autumn in SE France is favoured by the moist and unstable low-level jet from the Mediterranean, and triggered by steep mountain range (the Cévennes). This mechanism is crucial for the development of the Mesoscale Convective System (MCS) leading to extreme precipitation over this area (Ducrocq et al., 2008; Khodayar et al., 2016; Lee et al., 2018; Nuissier et al., 2008). However, the manifestation of the super-CC scaling has been observed in many other areas rather than the western Mediterranean where mechanisms leading to heavy convective precipitation events are different. The underlying theory to clarify this deviation from the Clausius-Clapeyron relation is still controversial. One can explain by the property of convective processes itself which is enhanced by the latent heat released during condensation as we discussed in line 234-235 of the preprint. However, the super-CC can also be explained by statistical effects at the transition-temperature range that convective and large-scale precipitation are combined (Berg and Haerter, 2013; Haerter and Berg, 2009; Molnar et al., 2015), or it can be the combination of both where MCS is embedded within a persistent large-scale frontal system and latent heat release favouring the moist updraft is involved in the MCS (Hatsuzuka et al., 2021). The appreciation of this mechanism is beyond the scope of our study and deserves further thorough investigation.

-L249 (Section 3.4): My understanding is that the analysis in this section is based on wet-events, i.e. days without precipitation are excluded. If this is the case, it would be useful for the reader to know what fraction of days contain wet events and if this differs much between the different simulations.

**Response:** We only consider wet-events (hourly/daily amount >= 0.1 mm) in our analysis. These wet-events account for 30% in observations dataset, from 40% to 50% in convection-permitting simulations (CPS) and from 50% to 60% in the EURO-CORDEX simulations. This also shows that the CPSs reduce the drizzle problem as stated in Kendon et al., (2012).

-L252: Change "either ... or" to "both ... and".

**Response:** We agree.

-Figure 3: There's no panel (i) after (h), so I think you need to change (j) to (i).
**Response:** Thank you for noticing this error.

-Figure 3: What does the yellow colour over Italy represent? If this is simply an area of no data, then it would be good to mask it in white like in Figure 2 (g).

**Response:** Yes, that is an area with no data. We will mask it.

Please pay attention that the Fig.1 is put at the end of this document.

**Reference**

Berg, P. and Haerter, J. O.: Unexpected increase in precipitation intensity with temperature — A
result of mixing of precipitation types?, Atmos. Res., 119, 56–61,
doi:https://doi.org/10.1016/j.atmosres.2011.05.012, 2013.

Ducrocq, V., Nuissier, O., Ricard, D., Lebeaupin, C. and Thouvenin, T.: A numerical study of
three catastrophic precipitating events over southern France. II: Mesoscale triggering and
stationarity factors, Q. J. R. Meteorol. Soc., 134(630), 131–145, doi:10.1002/qj.199, 2008.

Fumière, Q., Déqué, M., Nuissier, O., Somot, S., Alias, A., Caillaud, C., Laurantin, O. and Seity,
Y.: Extreme rainfall in Mediterranean France during the fall: added value of the CNRM-AROME
Convection-Permitting Regional Climate Model, Clim. Dyn., 1–15, 2019.

Haerter, J. O. and Berg, P.: Unexpected rise in extreme precipitation caused by a shift in rain type?,
Nat. Geosci., 2(6), 372–373, doi:10.1038/ngeo523, 2009.

Hardwick Jones, R., Westra, S. and Sharma, A.: Observed relationships between extreme sub-daily
precipitation, surface temperature, and relative humidity, Geophys. Res. Lett., 37(22),
doi:10.1029/2010GL045081, 2010.

Hatsuzuka, D., Sato, T. and Higuchi, Y.: Sharp rises in large-scale, long-duration precipitation
extremes with higher temperatures over Japan, npj Clim. Atmos. Sci., 4(1), 29,
doi:10.1038/s41612-021-00184-9, 2021.

Kendon, E. J., Roberts, N. M., Senior, C. A. and Roberts, M. J.: Realism of rainfall in a very high-
resolution regional climate model, J. Clim., 25(17), 5791–5806, 2012.

Khodayar, S., Fosser, G., Berthou, S., Davolio, S., Drobinski, P., Ducrocq, V., Ferretti, R., Nuret,
M., Pichelli, E., Richard, E. and Bock, O.: A seamless weather–climate multi-model
intercomparison on the representation of a high impact weather event in the western
Mediterranean: HyMeX IOP12, Q. J. R. Meteorol. Soc., 142(S1), 433–452,
doi:https://doi.org/10.1002/qj.2700, 2016.

Lee, K. O., Flamant, C., Duffourg, F., Ducrocq, V. and Chaboureau, J. P.: Impact of upstream
moisture structure on a back-building convective precipitation system in south-eastern France during HyMeX IOP13, Atmos. Chem. Phys., 18(23), 16845–16862, doi:10.5194/acp-18-16845-2018, 2018.

Molnar, P., Fatichi, S., Gaál, L., Szolgay, J. and Burlando, P.: Storm type effects on super Clausius–Clapeyron scaling of intense rainstorm properties with air temperature, Hydrol. Earth Syst. Sci., 19(4), 1753–1766, doi:10.5194/hess-19-1753-2015, 2015.

Nuissier, O., Ducrocq, V., Ricard, D., Lebeaupin, C. and Anquetin, S.: A numerical study of three catastrophic precipitating events over southern France. I: Numerical framework and synoptic ingredients, Q. J. R. Meteorol. Soc., 134(630), 111–130, 2008.

Prein, A. F., Rasmussen, R., Castro, C. L., Dai, A. and Minder, J.: Special issue: Advances in convection-permitting climate modeling, Clim. Dyn., 55(1), 1–2, doi:10.1007/s00382-020-05240-3, 2020.

[Figure]

**Fig.1: Rx1day (upper panel) and Rx3hour (lower panel) from CPSs upscaled to EUR-11 resolution. From left to right showing downscaling experiments from IPSL-CM5A-MR, HadGEM2-ES and NorESM1-M. Note that the color bar for Rx1day shows a different scale from one for Rx3hour.**

---

## Author Comment (AC3) · 7 Jun 2021

This manuscript describes a number of long-term simulations with a convective-permitting model over southern France focussing on its ability to reproduce extreme precipitation events in fall. The study, in which several model runs covering a large number of years with forcing data from different global climate models, is clearly interesting and results indicate that this model is more suited to simulate such events compared to standard high-resolution RCMs. As such studies are sparse this one can be an important contribution that merits publication.

**Response:** Thank you for the thorough review of our manuscript. We appreciate all the constructive comments and suggestions that you provide in your review, and that our work is valued by the reviewer.

However, the manuscript first needs some improvement. This is partly related to the language, I would suggest a thorough language check before re-submission. It can also be clearer what has been done (exactly which model version has been run?) and why (the choice of time periods including the mismatch with the observations used for evaluation). It is also not clearly explained how this high-resolution convection-permitting model performs when driven by perfect boundary conditions. As it is now, it is not clear if biases are related to poor forcing conditions (including wind, stability, SSTs) from the GCMs or if they result from poor model performance. The same is true for the coarser-scale EUR-11. The results in some of the figures are not entirely in line with how the text describes them.

**Response:** We will revise the text carefully before the re-submission.

We have mentioned in our preprint line 81 that the WRF-ARW version 3.8.1 is used to perform downscaling experiments in our study.

- We selected the two periods of 1951-1980 and 2001-2030 for our study because of two main reasons: 1) the forcing EURO-CORDEX of our simulations started in 1951, so we cannot go further past. 2) We would maximize the difference in Greenhouse gas concentrations between the two periods while the simulations are not going two far to the uncertain future scenarios for which we do not have the observations.

- For the mismatch between simulations and the benchmarks, we will clarify in your specific comment.

- As we stated in line 89 in our preprint that we will use the simulations generated in this study for further anthropogenic climate change impact investigation, we decided to perform our downscaling for EURO-CORDEX/CMIP5 runs rather than forcing our runs by perfect boundary conditions. In fact, we have done a few short runs (3 months) forced by ERA-Interim for the purpose of testing our CPS domain position. These results were provided in the supplementary section. We also keep in mind that several studies that we mentioned in our introduction had done convection-permitting simulations forced by ERA-Interim using different RCMs and focusing on different areas. And they found the advantages of this approach in replicating extreme precipitation events. Their findings fed our idea that we can step further in this field of modelling by running CPSs in climate scale and forced by CMIP5 boundaries. We will discuss our experiments with reanalysis forcing in the text.

- We will discuss the performance of forcing GCMs (especially SSTs) in another specific comment below.

- We will check and correct where the text does not describe the Figures correctly.

As for the structure of the paper, I find that there is no proper discussion of the results. Currently, there are some references alluded to and compared with both in the result section and in the conclusion chapter. I think that the discussion should go into either the results section or be introduced in a separate chapter of its own. Furthermore, the supplementary material is interesting and I'm thinking that it may be useful to include directly in the paper instead (it could be part of the discussion), the paper is not that extensive in its present form.

**Response:** We will discuss our results and tie the results to literature in the results and discussion section. For the supplementary section, we prefer to keep those materials in the supplementary. Because all figures were provided in similar styles as those in the main manuscript that may confuse the reader if we mix the two parts together. But as mentioned above, we will discuss the results from experiments forced by reanalysis on the main text of the revised manuscript.

**Specific comments:**

**RC:** Line 22 Please explain what is meant by "cloud-resolving". Most convective clouds are smaller than 3x3 km and are definitely not resolved by the convective-permitting models used here.

**Response:** Our simulations are "convection-permitting". However, in line 22, we mentioned "cloud-resolving" simulations in a general context. This was not implied for our model.

**RC:** Line 31-37 Now also shown for higher latitudes in Scandinavia (Lind et al., 2020, see https://link.springer.com/article/10.1007/s00382-020-05359-3)

**Response:** Thank you for mentioning those new interesting results. We will add this to the reference.

**RC:** Line 42 I would not use "large" and "robust" here to describe the number of simulations done and the status of knowledge. The number is in fact highly limited and only for a few regions mainly covering parts of the mid-latitudes.

**Response:** We replace "a large number of" by "a few" and remove "robust" from the text.

**RC:** Line 56-57 Instead of referring to a project (HyMex) I think it is more interesting for a reader to learn stg on what scientific questions are being addressed and/or why this is interesting from a societal perspective (the references given may be good here but I don't see the need for introducing the project).

**Response:** We will replace a sentence starting in line 55 by 3 sentences providing the motivation why the Mediterraen region has been receiving more interest and specific scientific questions are being addressed by research communities.
"*The coastal regions along the Mediterranean frequently undergo heavy precipitation events in*
*the autumn, which lead to flash floods and landslides causing massive losses and damages (Delrieu et al., 2005; Fresnay et al., 2012; Llasat et al., 2013; Nuissier et al., 2008; Ricard et al., 2012). In addition, this area is considered as a hotspot of climate change that strongly responds to warming at global scale (Giorgi, 2006; Tuel and Eltahir, 2020). As a result, the Mediterranean has received an increasing scientific interest in understanding mechanisms leading to flood-inducing heavy*
*precipitation as well as in improving the model ability to predict and project those events in a complex changing climate that provides substantial support to adaptation and mitigation for society (Drobinski et al., 2014; Ducrocq et al., 2014).*"

**RC:** Line 87-88 This is unclear. Why is the model run for 1951-1980 and not 1961-1990? Later
results are compared to observations from 1961-1990 and even if the results are from GCM-driven simulations there are forcing differences between these periods potentially compromising the comparison. This should be addressed in the paper. Furthermore, the use of 2001-2030, that is mostly based on a future scenario (RCP8.5) is also not clearly explained and in a way difficult to understand. In a similar way, comparison is done between observations covering 1998-2017 and
model simulations covering 2001-2030. Again, there is a mismatch of more than 10 years in a period with a strong global mean change. Is there any implication for the results from this (mismatch in extremes as simulated in the 2020ies with stronger forcing compared to the previous 20 years)?

**Response:** All simulations in this study are designed for evaluation of convection-permitting setup and further investigation of impact of human-induced climate change on current climate (i.e. 2001-2030) and historical climate (i.e. 1951-1980). For the latter purpose, we aim at maximizing the climate signal by selecting the two periods with the distance as large as possible. This explains why we selected 1951-1980 as a historical climate.

We then evaluate simulated daily precipitation indices of 1951-1980 against in situ observations and SAFRAN for 1961-1990. Our observations contain a few stations starting in 1951, while other stations start a few years later. In order to make a homogeneous length among stations and SAFRAN, given that SAFRAN starts in 1961, we select the 30 years period of 1961-1990 as a benchmark.

We also compare 3-hourly precipitation indices from 30 years (2001-2030) of simulations against 21 years (1998-2018) from in situ observations. For this set of observations, we only have those 21 years, therefore we cannot use them to evaluate our historical simulations. Note that our daily observations and 3-hour observations are two separate datasets.

**RC:** Line 91 Here, Cévennes is mentioned for the first time. For the not-so-very-French reader it is not clear where these mountains are. It becomes clearer when looking at subsequent figures. But, it would be good already in Figure 1 to illustrate where these mountains are (as part of the Central Massif – I guess?). Also the "Cevennes-box" could be given there. (Reference could also be given to this figure on line 141 where the box is detailed in the text).

**Response:** In the preprint, the Cévennes is mentioned for the first time in line 76. Indeed this mountain range is the southern part of the Massif Central in the south of France. We will state more clearly in the text and highlight it, and the Cévennes box Figure 1.

**RC:** Line 100 What is "the French Mediterranean Sea"? Is this stg outside of the territorial 12 nm zone?

**Response:** We mean our domain covers a large part of the Mediterranean that adjoins the French coast. We will change those to "*the French Mediterranean region*".

**RC:** Line 105-109 It is not clearly described in the paper how the current configuration of WRF 3.8.1 performs w.r.t. the observed precipitation extremes in any ERA-Interim driven simulation.

The supplementary material holds such ERA-Interim driven simulations, however, it should be better addressed at some point in the paper how this model (and the currently used setup) works. Also, why are these particular schemes mentioned here and not others? Is it clear from reading these few lines exactly which version of the model that has been used? Could someone else reproduce your experiment based on what is written here? On line 106 it says SSTs are updated every 6 hours. Is this also true for the lateral boundary conditions?

**Response: 1)** In the simulations at climate scale, we used the configuration similar to what was used in experiments driven by ERA-Interim. We will mention this in the main text of the manuscript. **2)** We mentioned in the text those schemes that have direct impacts on the development processes of precipitation and temperature. Those schemes are consistent with what were used in EURO-CORDEX. **3)** We mentioned clearly in line 81 that we used the WRF-ARW version 3.8.1. The simulations generated in this study can certainly be reproduced based on the information given in the text and additional configuration information of WRF that was used in the EURO-CORDEX experiments (Coppola et al., 2020; Vautard et al., 2020). **4)** We update the SSTs every day, which is consistent with the EURO-CORDEX experiments using WRF model. We will correct this information in the main text.

**RC:** Figure 1 Why is there no altitude associated with Menorca and Ibiza on the map. Are these islands not resolved by the model?

**Response:** Those islands are not represented in topo data of WRF.

**RC:** Line 146 Unclear what "few" means here. Is it only a few time steps from the 6480 time steps (27 hours times 60 minutes times 4 time steps per minute)? Or do you mean that the 6480 time steps are few relative to the full length of the simulations?

**Response:** We meant a few time steps with an interval of 3 hours for model simulations and 6 hours for ERA5. We will clarify in the article.

**RC:** Line 157 A reference is missing for ERA5.

Response: Added

**RC:** Line 227-240 I don't fully agree on the interpretation of the figures including the temperature intervals given in the text here. For instance, in Fig 4a I think it is quite clear that the approximate

CC-scaling holds between 3-13C. Between 13 and 18 there seems to be no such relation, but rather a constant precipitation rate regardless of temperature. Similarly, for 4b I think the super-CC scaling applies up to approximately 13C whereafter CC-scaling applies. Furthermore, it is not clear that the models reproduce the behaviour, in some aspects yes but not in the details. The slopes do differ. Also, the slopes differ between EUR-11 and CPS model versions (e.g. HadGEM). This text needs revision. I also think it would be easier to follow if the figure was remade so that the corresponding EUR-11 and CPS simulations (driven by the same GCM) where colored in the same way (suggested to be denoted with full and dashed lines).

**Response:** We have updated our routine in this scaling analysis by adding a threshold of at least 300 data points to take a bin into consideration. By doing so, a few bins in the lowest and highest temperature ranges were eliminated, therefore we can avoid the artificial effect of under-sampling. We also adjusted colors and lifestyle following the suggestion of the reviewer. The updated analysis shows that observed scaling follows the C-C relation in a range of 2°C to 13°C for daily precipitation (see Fig.1 at the end of this document by which the Figure 4 in the preprint will be replaced). The behavior of each convection-permitting simulation replicates its driving EURO-CORDEX model for the daily precipitation scaling analysis. Specifically, the 2 downscaling simulations of the IPSL-CM5A-MR reproduce roughly the C-C relation in a range of 9°C to 17°C, while the 2 downscaling simulations of the HadGEM2-ES follow the C-C in range of 5°C to 13°C. The 2 simulations of NorESM1-M show similar behaviour that follows the C-C in the range of 4°C to 14°C. The overall scaling rate from EUR-11 simulations are close to observations, while CPSs slightly overestimate this rate. For sub-daily precipitation scaling with temperature analyses, observations show a super C-C relation in the temperature range of 6°C to 13°C. The 3 CPSs can reproduce this feature, while the 3 EUR-11s completely fail to provide both super C-C and C-C relations. Specifically, CPS_IPSL-CM5A-MR shows a super C-C scaling in the range of 9°C to 17°C. The CPS_HadGEM2-ES and CPS_NorESM1-M follow super C-C in the range of 5°C to 17°C and 7°C to 14°C, respectively.

**RC:** Line 243-244 Here it says that "We could explain ... underestimation by the fact … simplified cloud process". I don't see how this is explained here! Please be more explicit.

**Response:** We will modify that sentence by " *... underestimation by the fact that the resolution of EUR-11 is insufficient to reproduce the more localized extreme events and* that the convection scheme … cloud process *by statistical distributions and imposing assumptions of quasi-equilibrium with large-scale forcing (from grid points), approximation of moist air entraining in the updraft, and representation of all single cloud elements by sole steady state updraft of the*

*whole cloud ensemble (Houze, 2004; Lenderink and Attema, 2015; Prein et al., 2013; de Rooy et al., 2013)".*

**RC:** Line 262 This section about moisture sources would be a good place to say stg more explicit about the underlying GCMs. From the figure it appears that the Hadley model have a better
representation of the moisture flux over the southern parts of the Mediterranean in association to the events examined. Another feature that could be addressed would be SSTs of the GCMs in association with the events. If some of them have strong biases it would likely influence the moisture source and transport. The moisture supply from the sea is of course very important in this aspect (as also shown in a convective-permitting model for this area by Lenderink et al., 2019,
https://iopscience.iop.org/article/10.1088/1748-9326/ab214a/pdf)

**Response:** There are many processes potentially contributing to the better moisture fluxes of the downscaling experiments, including dynamics and sea surface temperature. Here we will simply check the GCM SST biases with respect to the ERA5 for the 12 heaviest precipitation events. Fig.2
of this document shows that the IPSL-CM5A-MR and HadGEM2-es provide warm bias with their mean biases of 0.3 and 0.9°C, respectively, while the NorESM1-M gives cold bias of 2.2°C. This can partly explain why the downscaling experiments from NorESM1-M reproduce extreme rainfall over the Cévennes lower than others.

**Detailed minor comments:**
**RC:** In general, the manuscript needs a careful revision of the language. There are many examples of errors and/or things that could be clarified, some of which are given below.
**RC:** Line 8 Change "downscaled" to "run"
Response: OK.
**RC:** Line 10 Remove the first "simulations"

**Response:** OK.

**RC:** Line 24 Change into "is also hope"

**Response:** OK.

**RC:** Line 25 Consider changing to "conducting several runs to generate large ensembles of
simulations with sufficient resolution"

**Response:** OK.

**RC:** Line 45 Instead of "surface field" I would suggest "surface properties"

**Response:** We agree.

**RC:** Line 49 Remove "in the simulation results"

**Response:** OK.

**RC:** Line 59 Here is an example of a language problem where it says "Z et al found that convective-permitting model outperformed ...". Either it should be "a convection-permitting model" or "convection-permitting models".

**Response:** We do not see this error 'convective-permitting' anywhere in the preprint version.

**RC:** Line 72-73 This is difficult to understand. The "analysis" is not "downscaling results of EURO-CORDEX" as it says. Rather the analysis is undertaken on results from downscaling EURO-CORDEX simulations.

**Response:** We meant that the analysis in this article is made by downscaling "the existing" results of EURO-CORDEX experiments. To avoid misunderstanding, we will change that sentence to "The analysis made here is at a climate scale and is done by *dynamically downscaling the climate information provided by the existing EURO-CORDEX experiments*."

**RC:** Line 77-78 This is not really needed in this short paper that is quite standard in its structure. In case it is retained it could be explicitly mentioned that there is also supporting material and what can be found there (and why not in the paper itself?)

**Response:** We prefer keeping those sentences and will mention in the revised manuscript that the experiments done with WRF driven by ERA-Interim are provided in the supplementary material. We will also discuss the result from those experiments in the main text.

**RC:** Line 84-85 Shorter with ": : :EURO-11 simulations were also done with WRF-ARW version 3.8.1 driven by three general circulation models (GCMs): : :"

**Response:** OK. The sentence is rewritten as "*These EUR-11 simulations were also done with WRF-ARW version 3.8.1 and driven by three General Circulation Models (GCMs) including the*
*IPSL....*"

**RC:** Line 128 It is not the "moisture sources" but the "moisture" that is transported from the Med Sea. And on the same line not the "massive moisture" but the "massive amount of moisture".

**Response:** We will replace the former by "*moisture*" and the latter by "*massive amount of moisture*" following the suggestions of the reviewer.

**RC:** Line 136 Suggest to replace "zonal and meridional" with "horizontal"

**Response:** We agree.

**RC:** Line 137 "hPa" instead of "mb"

**Response:** We agree.

**RC:** Line 251-252 I think the () can be removed here. References to the figures are given appropriately in the subsequent text.

**Response:** We agree

**RC:** Line 255 Should it be "-45%" here instead of 40?

**Response:** We agree

**RC:** Line 263 It is not the ability of the "simulation" but of the "model" that is investigated.

**Response:** We agree

Please pay attention that the Fig.1 and 2 are put at the end of this document.

**Reference**

Coppola, E., Nogherotto, R., Ciarlò, J. M., Giorgi, F., van Meijgaard, E., Iles, C., Kadygrov, N., L. Corre, M. S., Somot, S., Nabat, P., Vautard, R., Levavasseur, G., Schwingshackl, C., Sillmann, J., Kjellström, E., Nikulin, G., Aalbers, E., Lenderink, G., Christensen, O. B., Boberg, F., Sørland, S. L., Demory, M.-E., Bülow, K. and Teichmann, C.: Assessment of the European climate projections as simulated by the large EURO-CORDEX regional climate model ensemble, J. Geophys. Res. sub judice, 2020.

Delrieu, G., Nicol, J., Yates, E., Kirstetter, P.-E., Creutin, J.-D., Anquetin, S., Obled, C., Saulnier, G.-M., Ducrocq, V., Gaume, E., Payrastre, O., Andrieu, H., Ayral, P.-A., Bouvier, C., Neppel, L., Livet, M., Lang, M., du-Châtelet, J. P., Walpersdorf, A. and Wobrock, W.: The Catastrophic Flash-Flood Event of 8–9 September 2002 in the Gard Region, France: A First Case Study for the Cévennes–Vivarais Mediterranean Hydrometeorological Observatory, J. Hydrometeorol., 6(1), 34–52, doi:10.1175/jhm-400.1, 2005.

Drobinski, P., Ducrocq, V., Alpert, P., Anagnostou, E., Béranger, K., Borga, M., Braud, I., Chanzy, A., Davolio, S., Delrieu, G., Estournel, C., Boubrahmi, N. F., Font, J., Grubišić, V., Gualdi, S., Homar, V., Ivančan-Picek, B., Kottmeier, C., Kotroni, V., Lagouvardos, K., Lionello, P., Llasat, M. C., Ludwig, W., Lutoff, C., Mariotti, A., Richard, E., Romero, R., Rotunno, R., Roussot, O., Ruin, I., Somot, S., Taupier-Letage, I., Tintore, J., Uijlenhoet, R. and Wernli, H.: HyMeX: A 10-Year Multidisciplinary Program on the Mediterranean Water Cycle, Bull. Am. Meteorol. Soc., 95(7), 1063–1082, doi:10.1175/bams-d-12-00242.1, 2014.

Ducrocq, V., Braud, I., Davolio, S., Ferretti, R., Flamant, C., Jansa, A., Kalthoff, N., Richard, E., Taupier-Letage, I., Ayral, P.-A., Belamari, S., Berne, A., Borga, M., Boudevillain, B., Bock, O., Boichard, J.-L., Bouin, M.-N., Bousquet, O., Bouvier, C., Chiggiato, J., Cimini, D., Corsmeier, U., Coppola, L., Cocquerez, P., Defer, E., Delanoë, J., Girolamo, P. Di, Doerenbecher, A., Drobinski, P., Dufournet, Y., Fourrié, N., Gourley, J. J., Labatut, L., Lambert, D., Coz, J. Le, Marzano, F. S., Molinié, G., Montani, A., Nord, G., Nuret, M., Ramage, K., Rison, W., Roussot, O., Said, F., Schwarzenboeck, A., Testor, P., Baelen, J. Van, Vincendon, B., Aran, M. and Tamayo, J.: HyMeX-SOP1: The Field Campaign Dedicated to Heavy Precipitation and Flash Flooding in the Northwestern Mediterranean, Bull. Am. Meteorol. Soc., 95(7), 1083–1100, doi:10.1175/bams-d-12-00244.1, 2014.

Fresnay, S., Hally, A., Garnaud, C., Richard, E. and Lambert, D.: Heavy precipitation events in the Mediterranean: sensitivity to cloud physics parameterisation uncertainties, Nat. Hazards Earth

Syst. Sci., 12(8), 2012.

Giorgi, F.: Climate change hot-spots, Geophys. Res. Lett., 33(8), doi:10.1029/2006gl025734, 2006.

Houze, R. A.: Mesoscale convective systems, Rev. Geophys., 42(4), 1–43, doi:10.1029/2004RG000150, 2004.

Lenderink, G. and Attema, J.: A simple scaling approach to produce climate scenarios of local precipitation extremes for the Netherlands, Environ. Res. Lett., 10(8), 85001, 2015.

Llasat, M. C., Llasat-Botija, M., Petrucci, O., Pasqua, A. A., Rosselló, J., Vinet, F. and Boissier, L.: Towards a database on societal impact of Mediterranean floods within the framework of the
HYMEX project, Nat. Hazards Earth Syst. Sci., 13(5), 1337–1350, doi:10.5194/nhess-13-1337-2013, 2013.

Nuissier, O., Ducrocq, V., Ricard, D., Lebeaupin, C. and Anquetin, S.: A numerical study of three catastrophic precipitating events over southern France. I: Numerical framework and synoptic
ingredients, Q. J. R. Meteorol. Soc., 134(630), 111–130, 2008.

Prein, Gobiet, A., Suklitsch, M., Truhetz, H., Awan, N. K., Keuler, K. and Georgievski, G.: Added value of convection permitting seasonal simulations, Clim. Dyn., 41(9–10), 2655–2677, 2013.

Ricard, D., Ducrocq, V. and Auger, L.: A Climatology of the Mesoscale Environment Associated with Heavily Precipitating Events over a Northwestern Mediterranean Area, J. Appl. Meteorol. Climatol., 51(3), 468–488, doi:10.1175/JAMC-D-11-017.1, 2012.

de Rooy, W. C., Bechtold, P., Fröhlich, K., Hohenegger, C., Jonker, H., Mironov, D., Pier
Siebesma, A., Teixeira, J. and Yano, J.-I.: Entrainment and detrainment in cumulus convection: an overview, Q. J. R. Meteorol. Soc., 139(670), 1–19, doi:https://doi.org/10.1002/qj.1959, 2013.

Tuel, A. and Eltahir, E. A. B.: Why Is the Mediterranean a Climate Change Hot Spot?, J. Clim., 33(14), 5829–5843, doi:10.1175/JCLI-D-19-0910.1, 2020.

Vautard, R., Kadygrov, N., Iles, C., Boberg, F., Buonomo, E., Bülow, K., Coppola, E., Corre, L., van Meijgaard, E., Nogherotto, R., Sandstad, M., Schwingshackl, C., Somot, S., Aalbers, E.,

Christensen, O. B., Ciarlo`, J. M., Demory, M.-E., Giorgi, F., Jacob, D., Jones, R. G., Keuler, K., Kjellström, E., Lenderink, G., Levavasseur, G., Nikulin, G., Sillmann, J., Sørland, S. L., Solidoro, 395 C., Steger, C., Teichmann, C., Warrach-Sagi, K. and Wulfmeyer, V.: Evaluation of the large EURO-CORDEX regional climate model ensemble, J. Geophys. Res. sub judice, 2020.

[Figure]

**Fig. 1 : Extreme (99th percentile) daily precipitation (a) and daily maximum of 3-hourly rainfall (b) in scaling with daily temperature at 2m from simulations (1951-1980 for daily rainfall and 2001-2030 for 3-hourly rainfall) and in situ observations (1961-1990 for daily rainfall and 1998-2018 for 3-hourly rainfall); the black dot lines show Clausius-Clapeyron relation and the red dot lines show the super Clausius-Clapeyron relation; the grey band denotes 90% confident interval of observational scaling.**

[Figure]

**Fig. 2 : Differences in sea surface temperature averaged over the 12 events from the 3 forcing GCMs with respect to the ERA5.**

---

## Author Response (AR2)

**Review of revised manuscript "Evaluation of convectionpermitting extreme precipitation simulations for the south of France" by Luu et al. (2021)**

5 The revised manuscript represents a faithful response to most of the comments. The issue of comparing model and observational data at different spatial scales remains unresolved.

To recap, my previous criticism was that the authors were directly comparing results from the 12 km and 3 km resolution models with station data and 1 km resolution observations, before
judging which model performs better. I argued that this is not a fair way to judge added value as the 12 km model is designed to represent grid box means at the 12 km scale, not values at the point (station data) or 1 km (gridded observations) scale. Closer agreement of the 3 km model with the aforementioned observations does not, therefore, necessarily mean that the 3 km model is "better" than the 12 km model, but rather likely simply reflects that the observations'
resolution are closer to that of the 3 km model. My argument was that to identify if the 3 km

model "adds value" to the 12 km model, one must first upscale the model and observational data to the resolution of the coarsest data (in this case, the 12 km model).

Response: We thank the reviewer for his/her enthusiasm in reviewing our revised manuscriptand providing further discussions on the "change of support" issue.

**Main comments.**

The authors disagreed with my above criticism. Their arguments are summarized below in 25 "Authors C1-4". My responses follow in "Reviewer C1-4".

\*Authors C1. Model users regularly use coarse-resolution data (e.g. 5 to 50 km) for local climate studies. The 3 km model's higher spatial variability and improved precipitation at small scales thus represent added value for these users. The authors only focus on to what extent the 3 km model improves the extreme precipitation at local scale.

30

\*\*Reviewer C1. It is true that some users directly use low-resolution climate data for point- or local-scale studies. This, however, does not mean that those users are correct to do so and is, anyway, of secondary importance to my criticism. If the stated aim of the research is

- 35 "evaluation" and to "investigate the added value" (see title/abstract), then the fact remains that it is not appropriate to do this at a spatial scale that the 12 km model is not intended to represent. It is trivial that the 3 km model exhibits higher spatial variability (simply because it has more grid cells); added detail is not added value.
- 40 The further the model resolution is away from the observation's resolution, then the less appropriate the comparison. Hence, if the 3 km and 12 km models were to be perfect at their own spatial scales, then the 3 km model must be in better agreement with the point- and kilometre-scale observations, compared to the 12 km model. This does not mean that the 3 km model "adds value"; it simply reflects the different scales the models are intended to represent.
- 45

In short, it is not possible to make conclusions on added value if the two models are being compared at different spatial scales.

Response: In order to broaden the discussion and include the reviewer's viewpoint, in this revision, we also upscaled (with some modification in upscaling procedure compared to our previous revision) our convection permitting simulations to 12 km as in EUR-11 simulations and present both comparisons. The results allow an interesting comparison which is presented in the response beneath.

- \*Authors C2. Their goal is to "assess the overall improvement against observed station data", not to disentangle the causes of 3 km model improvement, i.e. resolution or physics. Comparing the 12 km and 3 km models at the same resolution (i.e. 12 km) would only answer whether (or why) the fine-scale resolution (3 km) can improve the larger scale (12 km).
- 60 \*\*Reviewer C2. I accept that disentangling the contributions of different resolution and physics to any added value is not the aim of the study, so no problems there. I also agree that comparing the 3 km and 12 km models at the same (12 km) resolution will "only" answer whether the fine-scale resolution adds value at the 12 km scale. For the reasons outlined above, this (12 km) is however the minimum scale at which you can assess the added value.
- 65

Response: We now include both comparisons so the discussion can be broadened.

\*Authors C3. There is no "standard" way of evaluating model added value: the appropriate method depends on the scientific question.

70

\*\*Reviewer C3. I agree, but the scientific question also has to be appropriate. Asking whether 3 km simulations add value over 12 km simulations for representing point- or kilometre-scale observations (without upscaling) is, in my view, not an appropriate scientific question.

**75 **Response**: We understand the viewpoint, and include the proposed comparison with upscaled results, along with a direct comparison.**

\*Authors C4. The authors provide an additional analysis in their response where they upscale the 3 km data to the 12 km grid (observations are not upscaled) and re-compare the seasonal maxima (3 h, 1 day) against observations (1 km and stations). Based on this comparison, the 3 km model

\*\*Reviewer C4. I would like to know how the authors performed the upscaling for the results shown in Figure 4 of the response (this was unfortunately not mentioned). I ask this because the boxmean values for CPS (Figures 2/3, manuscript) and CPS-11 (Figure 4, response) are identical

within a rounding error of 0.1 mm, which seems highly implausible.

is deemed to still outperform the 12 km data.

80

85

The correct way to do the upscaling would be to upscale all the 3 hourly (daily) data to the EUR-11 grid, and then \*after\* that compute the Rx3hour (Rx1day) values. In Figure 4 it looks like the 90 upscaling has simply been performed on the final Rx3hour (Rx1day) results of the original CPS grid. What else could explain the identical boxmean values between CPS and CPS-11?

Response: In our previous revision, we indeed upscaled the final results of Rx1day and Rx3hour using conservative remap method provided in CDO. So, the identical box mean values between
95 CPS and CPS-11 is understandable because the conservative remap is expected to retain the flux of a variable over the domain. This could be the explanation for the approximation of box means between the CPS and CPS-11 (i.e., the CPS upscaled to 0.11 degree). However, as proposed by

the reviewer, we applied this upscaling procedure to every single field of daily rainfall and daily

- maximum 3-hourly rainfall from the CPSs, then we calculated Rx3hour and Rx1day again. We
  also upscaled the results from 11 years of COMEPHORE radar from 1 km to 12 km for Rx3hour as a reference to allow comparison. For Rx1day, we used SAFRAN (1961-1990) at its original resolution (8 km), which is, in our perspective, comparable to simulations at 12 km. The results are shown in Fig. 1 (for Rx3hour) and Fig. 2 (for Rx1day) below. Generally, the results from CPS-11 of this revision are 2% to 5% lower than the CPS-11 from our previous revision. For
- 105 Rx3hour, the mean of the Cévennes box from CPS-11 ranges from 1% to 15% lower than the result of upscaling COMEPHORE (Fig. 1). Meanwhile, the EUR-11 simulations (Figure 3a-c in

our previous revised manuscript) underestimate by 50% the mean of the Cévennes box in comparison with the upscaling COMEPHORE. For Rx1day (Fig. 2), the results from CPS-11 show biases of mean of Cévennes box ranging from -20% to 14% compared to SAFRAN (8 km).
110 While the EUR-11 simulations underestimate the mean of the Cévennes box by 20% to 40%. In summary, we find here that the upscaling procedure (to 12km) barely alters the results (5% max), while the differences between simulations are between 15% and 50%. Therefore the convection permitting model improves extreme precipitation simulation over the south of France. And as we stated above, we included both CPS and CPS-11 simulations and updated the text in our revised

---

## Author Response (AR3)

**Review of revised manuscript "Evaluation of convection-permitting extreme precipitation simulations for the south of France" by Luu et al. (2021)**

I have now the comments of reviewer #3. This reviewer thinks you improved the manuscript but has another point you should take care of.

**Response:** We thank you and the anonymous reviewer for your enthusiasm in the evaluation process of our study. Please find below our point-to-point responses to comments of the reviewer.

"The authors have addressed the aforementioned issue for Sections 3.1 and 3.2, where daily rainfall and daily maximum 3 h rainfall are compared (Figures 2 and 3) with the SAFRAN and COMEPHRE gridded products. This change is recognized. However, in Section 3.3 (Figures 4 and 5), the authors still compare EUR-11 data (0.11°) with CPS data (0.0275°) that has not been upscaled to the EUR-11 grid. On top of this, the authors are here comparing the aforementioned gridded data against point observations. As explained in previous reviews, I don't see the value of comparing precipitation intensities this way (e.g. 0.11° -vs- station data), particularly in the case of subdaily extreme precipitation. If the authors wish to persist with this, then they should also show the upscaled CPS-11 data in Figures 4 and 5, as they did in Figures 2 and 3, and make appropriate changes to the text in Section 3.3. That should be enough to finally get this paper over the line.

**Response:** We added the CPS-11 in our analyses of distribution of wet events (e.g., Figure 4 and 5). We also added a few changes to the text in Section 3.3 to discuss the added features from Figure 4 and 5.
* * *
Minor comments.

-L141: when discussing the impact of undersampling on the scaling curve, I suggest citing the study of Boessenkool et al. (2017) as a pointer to further information on the issue."

**Response:** We added the reference.